# Cellular and synaptic phenotypes lead to disrupted information processing in *Fmr1-KO* mouse layer 4 barrel cortex

Aleksander P.F. Domanski [1,2,3,4]*, Sam A. Booker[2,3,5], David J.A. Wyllie [2,3,5,6], John T.R. Isaac[4,7]* & Peter C. Kind[2,3,5,6]*

Sensory hypersensitivity is a common and debilitating feature of neurodevelopmental disorders such as Fragile X Syndrome (FXS). How developmental changes in neuronal function culminate in network dysfunction that underlies sensory hypersensitivities is unknown. By systematically studying cellular and synaptic properties of layer 4 neurons combined with cellular and network simulations, we explored how the array of phenotypes in *Fmr1*-knockout (KO) mice produce circuit pathology during development. We show that many of the cellular and synaptic pathologies in *Fmr1-KO* mice are antagonistic, mitigating circuit dysfunction, and hence may be compensatory to the primary pathology. Overall, the layer 4 network in the *Fmr1-KO* exhibits significant alterations in spike output in response to thalamocortical input and distorted sensory encoding. This developmental loss of layer 4 sensory encoding precision would contribute to subsequent developmental alterations in layer 4-to-layer 2/3 connectivity and plasticity observed in *Fmr1-KO* mice, and circuit dysfunction underlying sensory hypersensitivity.

[1] School of Physiology, Pharmacology & Neuroscience, University of Bristol, Bristol, UK. [2] Centre for Discovery Brain Sciences, University of Edinburgh, Hugh Robson Building, George Square, Edinburgh EH8 9XD, UK. [3] Patrick Wild Centre, University of Edinburgh, Hugh Robson Building, George Square, Edinburgh EH8 9XD, UK. [4] Developmental Synaptic Plasticity Section, NINDS, NIH, Bethesda, MD 20892, USA. [5] Simons Initiative for the Developing Brain, University of Edinburgh, Hugh Robson Building, George Square, Edinburgh EH8 9XD, UK. [6] Centre for Brain Development and Repair, NCBS, GKVK Campus, Bangalore 560065, India. [7]Present address: Janssen Neuroscience, J&J London Innovation Centre, J&J London Innovation Centre, One Chapel Place, London W1G 0B, UK. *email: aleks.domanski@bristol.ac.uk; jisaac5@ITS.JNJ.com; pkind@ed.ac.uk

ndividuals affected by many types of Autism Spectrum Disorder (ASD) and Intellectual Disabilities (ID) commonly exhibit sensory perceptual disturbances and tactile reactivity that span multiple modalities[1–3]. Fragile X syndrome (FXS) is a leading heritable cause of ASD/ID[4] with symptoms including seizures, tactile hypersensitivity and abnormal behaviours that affect early sensory and cognitive development. FXS is caused by loss of FMRP protein following transcriptional silencing of the Fmr1 gene. Like FXS, the Fmr1-KO mouse model[5] lacks FMRP and exhibits sensory, behavioural and cognitive deficits[6,7]. Sensory dysfunction in FXS and related ASDs have been proposed to underlie a range of behavioural and cognitive symptoms. In support of this hypothesis, a recent study has indicated a causal link between sensory dysfunction and social and repetitive behaviours in a mouse model of autism[8]. Hence a detailed understanding the sensory function in FXS may be critical to developing novel therapies.

Rodent models demonstrate that the sensory hypersensitivities associated with Fmr1 deletion are mirrored by an increase in circuit excitability[9–13]. However, numerous cellular processes contribute to circuit hyperexcitability in Fmr1-KO mice[14] and the potential number of mechanisms is even greater. FMRP has the potential to regulate the translation of diverse classes of neuronal mRNA[15–17] including ion channels, neurotransmitter receptor subunits, and intracellular signalling molecules. Furthermore, protein–protein interactions with voltage gated ion channels directly link FMRP to maintenance of intrinsic neuronal properties[11,18–20]. Finally alterations of the excitatory/inhibitory (E/I) balance, proposed to form a key component of circuit dysfunction in neurodevelopmental disorders[21,22] have been reported in Fmr1-KO mice[9,23–27] although the precise cellular mechanism underlying this E/I imbalance is unknown. Hence, a detailed dissection of the contribution of individual cellular phenotypes underlying the emergent circuit pathophysiology is required to understand the sensory processing deficits associated with FXS[10].

In the somatosensory cortex of Fmr1-KO mice, circuit dysfunction arises very early in development correlating with the peak of FMRP expression during the second postnatal week[28], a developmental stage marked by both the end of critical period refinement for thalamocortical (TC) synaptic input and the coordinated maturation of cortical layer 4 cell-intrinsic properties and recurrent network circuitry[29–33]. Loss of FMRP delays the onset and termination of the critical period for synaptic plasticity at TC synapses[28]. The consequences of this delay, both in terms of cellular and circuit function, are unknown, however, it is notable that active whisking begins soon after[34]. Hence, changes in the cellular and circuit physiology in layer 4 at this age could dramatically alter the nature of the sensory information being transmitted to layer 2/3 that drives further experience-dependent development. Importantly, at later ages Fmr1-KO mice display disrupted functional connectivity in layer 4 and an altered synaptic E/I balance[24,35–37]. However, whether these differences arise as a direct result of the loss of FMRP or are compensatory changes resulting from earlier developmental alterations in cellular physiology is not known. Furthermore, it is unclear if the cellular abnormalities that result from a delay in the sensitive period for synaptic plasticity at TC synapses underlies the altered E/I balance.

To address these questions, we examined the cellular and circuit properties of excitatory and feed-forward inhibition providing Fast-spiking GABAergic neurons (FS) in layer 4 of barrel cortex at postnatal days 10–11 (P10–11)[29,38,39], immediately after the termination of the delayed sensitive period for synaptic plasticity in Fmr1-KO mice and immediately prior to the onset of whisking behaviour. By combining brain slice electrophysiology and computational modelling, we show that an array of cellular-level pathologies is observed in Fmr1-KO layer 4: in connectivity, in cellular intrinsic properties and in synaptic function. The net result of these pathologies is a circuit with a lower threshold for action potential generation in response to TC input, but less well-timed firing relative to input stimuli. Our modelling shows that the cellular-level pathologies observed in the Fmr1-KO are often antagonistic in terms of circuit function suggesting that some 'pathologies' rather may be compensatory adaptations. Despite the compensation, the layer 4 circuit is dysfunctional in Fmr1-KO with a reduced ability to encode information manifesting as a reduction in pattern classification accuracy as relayed to layer 2/3.

## Results

**Altered membrane properties and excitability in _Fmr1-KO_ mice.** Previous work has identified a number of cellular, synaptic and circuit changes in Fmr1-KO mouse barrel cortex[9,24,40]. However, these experiments were conducted 5 days after the end of the delayed critical period for long-term potentiation (LTP) in Fmr1-KO mice. Therefore, it is unclear whether these changes are pathological or compensatory to network function or indeed how these cellular and synaptic mechanisms interact to produce circuit level deficits. To address this, we systematically analysed cellular, synaptic and network changes in thalamocortical brain slices[41,42] from younger Fmr1-KO mice immediately following the period of LTP at thalamocortical synapses and evaluated the effects of these mechanisms at a circuit level using computer simulations of layer 4 barrel cortex.

We first investigated whether there were changes in passive membrane properties in layer 4 barrel cortex neurons in acute slices from P10/11 Fmr1-KO mice compared with wild-type (WT) littermates, using whole-cell patch-clamp recordings. The principal cell type in layer 4 barrel cortex is the stellate cell (SC), which are recurrently connected glutamatergic neurons that project to layer 2/3[43–47]. Assessing intrinsic excitability, compared with WTs, Fmr1-KO SCs required less injected current to fire an action potential (lower rheobase) and exhibited an increase in both input resistance and membrane time constant (Fig. 1a), but no change in membrane capacitance or resting membrane potential (Fig. S1). SCs in Fmr1-KOs also had enhanced excitability with an increase in the number of action potentials elicited by 500 ms depolarising current steps (Fig. 1b); however, this increase in action potential number was associated with a decrease in action potential frequency during the early part of the depolarisation resulting in a reduction in action potential frequency adaptation, as assessed over a duration necessary to fire a standardised number of spikes (Fig. 1c). Moreover, in agreement with a previous finding in Fmr1-KO mouse hippocampus[20] action potential kinetics were slowed in Fmr1-KOs, with an increase in width and a decrease in amplitude (Fig. 1d).

A similar analysis was performed on the intrinsic excitability of fast-spiking interneurons. FS provide strong feed-forward inhibition (FFI) onto SCs and play a critical role in determining the integration of TC input and action potential output of SCs[30,31,38,48,49]. Similar to SCs, we observed an increase in input resistance and membrane time constant and reduced rheobase (Fig. 1e), and no change in whole-cell capacitance or resting membrane potential (Fig. S1b). However, in contrast to SCs, FS exhibited a reduction in action potential number during depolarisation (Fig. 1f) accompanied by a decrease in both action potential frequency and frequency accommodation (Fig. 1g) and a slowing of action potential kinetics (Fig. 1h, i).

Thus, in layer 4 of P10/11 Fmr1-KO mice both SCs and FS have altered passive membrane properties such that they produce action potentials in response to less depolarisation, compared

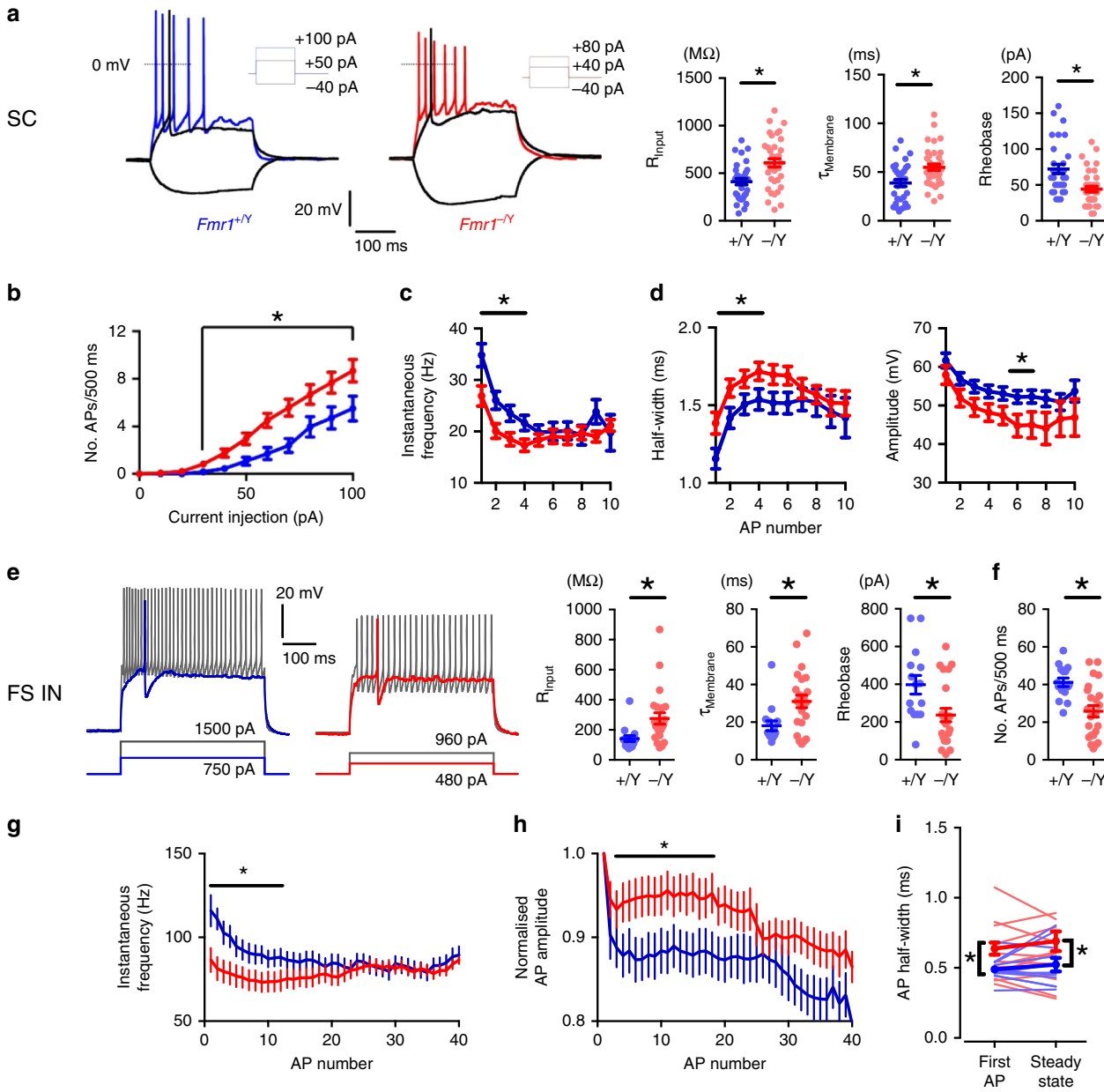

**Fig. 1** Altered intrinsic properties of *Fmr1-KO* Layer 4 SCs and FS interneurons. **a** Left: Example firing characteristics of layer 4 excitatory neurons in response to 500 ms hyper/depolarizing current injections (−40 pA, rheobase, 2× rheobase shown). Right: Passive membrane and intrinsic properties of L4 excitatory neurons: Input resistance (*Fmr1$^{+/Y}$*: 412 ± 33 MΩ, *Fmr1$^{-/Y}$*: 609 ± 43 MΩ, $p = 0.0007$), Membrane time constant (*Fmr1$^{+/Y}$*: 39 ± 3.7 ms, *Fmr1$^{-/Y}$*: 55 ± 3.2 ms, $p = 0.0015$), Rheobase current ($p < 0.002$, *Fmr1$^{+/Y}$*: 72 ± 6.6pA; $N = 33$, *Fmr1$^{-/Y}$*: 44 ± 4.0 pA; $N = 37$). Not shown: Membrane capacitance (*Fmr1$^{+/Y}$*: 94 ± 6.7 pF, *Fmr1$^{-/Y}$*: 89 ± 4.9 pF, $p = 0.56$), resting membrane potential (*Fmr1$^{+/Y}$*: −64 ± 1.7 mV, *Fmr1$^{-/Y}$*: −64 ± 1.3 mV, $p = 0.88$). All statistics herein: Student's *t*-test, two-tailed. **b** Suprathreshold current-spike frequency (FI) responses of L4 *Fmr1$^{-/Y}$* excitatory neurons were significantly steeper for current injections > 30 pA ($p = 0.02$, Mann–Whitney, *Fmr1$^{+/Y}$*: 120 ± 10 Hz/nA, *Fmr1$^{-/Y}$*: 200 ± 12 Hz/nA (*n*: 28 *Fmr1$^{+/Y}$*, 28 *Fmr1$^{-/Y}$*). **c** SC firing rate during twice-rheobase current injections. Asterisks: $p < 0.05$, *t*-tests comparing values for each spike position in train, N: *Fmr1$^{+/Y}$* = 50 neurons, *Fmr1$^{-/Y}$* = 42 neurons. **d** SC action potential half-width and amplitude during entrained firing to twice-rheobase current injections. Statistics as (**c**). **e** Left: Example spike waveforms fired by FS interneurons in response to 500 ms depolarizing current injections (rheobase, 2× rheobase shown). Right: passive membrane and intrinsic properties of FS interneurons (Asterisks: $p < 0.05$, *t*-test, N (neurons): *Fmr1$^{+/Y}$* = 15, *Fmr1$^{-/Y}$* = 23). **f–h** FS overall action potential rate (**f**), firing rate accommodation (**g**) and amplitude accommodation (**h**) during entrained firing to twice-rheobase current injections. Statistics as in (**e**). **i** Reduced rate of AP firing in *Fmr1$^{-/Y}$* FS interneurons as analysed by instantaneous frequency (1/inter-spike interval) of first and last two APs in train Light colours indicate individual neurons, thick bars/lines are mean ± SEM for each genotype. Asterisks on left and right graphs indicate parameters significantly different between genotypes compared by *t*-test ($p < 0.05$, N (neurons): *Fmr1$^{+/Y}$* = 15, *Fmr1$^{-/Y}$* = 27). Asterisks on centre graph indicate significantly different mean frequencies between genotypes compared by one-way ANOVA with Bonferroni's correction for multiple comparisons ($p < 0.05$, N: *Fmr1$^{+/Y}$* = 15, *Fmr1$^{-/Y}$* = 27)

with wild type. During sustained depolarisation, *Fmr1-KO* SCs continue to exhibit increased action potential output albeit with alterations in frequency and kinetics. In contrast, FS show decreased action potential output in response to sustained depolarisation as well as changes in frequency and kinetics.

**Increased response to depolarisation in SCs in *Fmr1-KO* mice.**
SCs are the output neurons from layer 4 and as such play an important role to integrate ascending TC input and provide output to layer 2/3, encoding stimulus features such as the presence of an object and its surface quality in spike frequency and times[38,39]. The capacity of neurons to faithfully follow rhythmic synaptic input, i.e. their impedance, is fundamentally constrained by their intrinsic properties[50]. We therefore next investigated whether the changes in intrinsic properties of SCs in *Fmr1-KO* mice alter their ability to transform inputs at ethologically relevant frequencies associated with somatosensation[51–55]. We assessed SC impedance properties[56,57], using a sinusoidal current of progressively increasing frequency and measured the resulting voltage response. SCs from *Fmr1-KO* mice exhibited an increased impedance between 0.5 and 4 Hz (Fig. S2a), consistent with predictions from the change in passive membrane properties (Fig. S3). This frequency-dependent response suggests that SCs in *Fmr1-KO* mice should exhibit alterations in action potential generation in response to membrane depolarisations of the same frequency range. To test this, we applied suprathreshold sinusoidal depolarisations at specific frequencies and found an increase in the number of action potentials elicited at low frequencies in *Fmr1-KO* compared with wild-type mice (0.5–4 Hz; Fig. S2B). Moreover, a significant phase shift in the timing of

action potential generation was also observed, for example at 10 Hz (Fig. S2C).

Thus, the alteration in the passive membrane properties of SCs in *Fmr1-KO* mice results in a selective increase in membrane depolarisation in response to low frequency depolarising currents. This produces both an increase in action potential output and a shift in action potential timing in response to low frequency depolarising currents.

**Altered feed-forward inhibition on SCs in *Fmr1-KO* mice.** FFI is a critical mechanism that governs the integration of TC input by SCs[30–32,48,58]. In layer 4, barrel cortex FFI is mediated by FS neurons which provide the TC-evoked inhibition onto SCs. This form of evoked E/I balance sets an integration window for SC action potential output between the onset of excitatory and the strong but delayed inhibitory inputs[48]. To determine if there are alterations in FFI in *Fmr1-KO* mice, we first investigated connectivity between FS and SCs using simultaneous whole-cell patch-clamp recordings. Similar to findings by Gibson et al. at P14[24], at P10/11, we found reductions in connectivity from SC to FS, but additionally from FS to SC and reciprocally between SC and FS, in *Fmr1-KO* mice compared with wild types (Fig. 2a). In contrast, however, the strength of the connections between FS and SCs was unchanged in *Fmr1-KO* mice compared with wild types (Fig. 2b) and the kinetics of the IPSCs onto SCs and of EPSCs into FS were also unchanged (Fig. 2c). Furthermore, there was no difference in SC–SC connectivity in *Fmr1-KO* mice compared with wild type, nor in synaptic strength or kinetics (Fig. 2d–f).

We next directly measured FFI in SCs, by stimulating TC inputs and comparing the size of the monosynaptic TC-evoked EPSC (at −70 mV holding potential) and the size of the di-

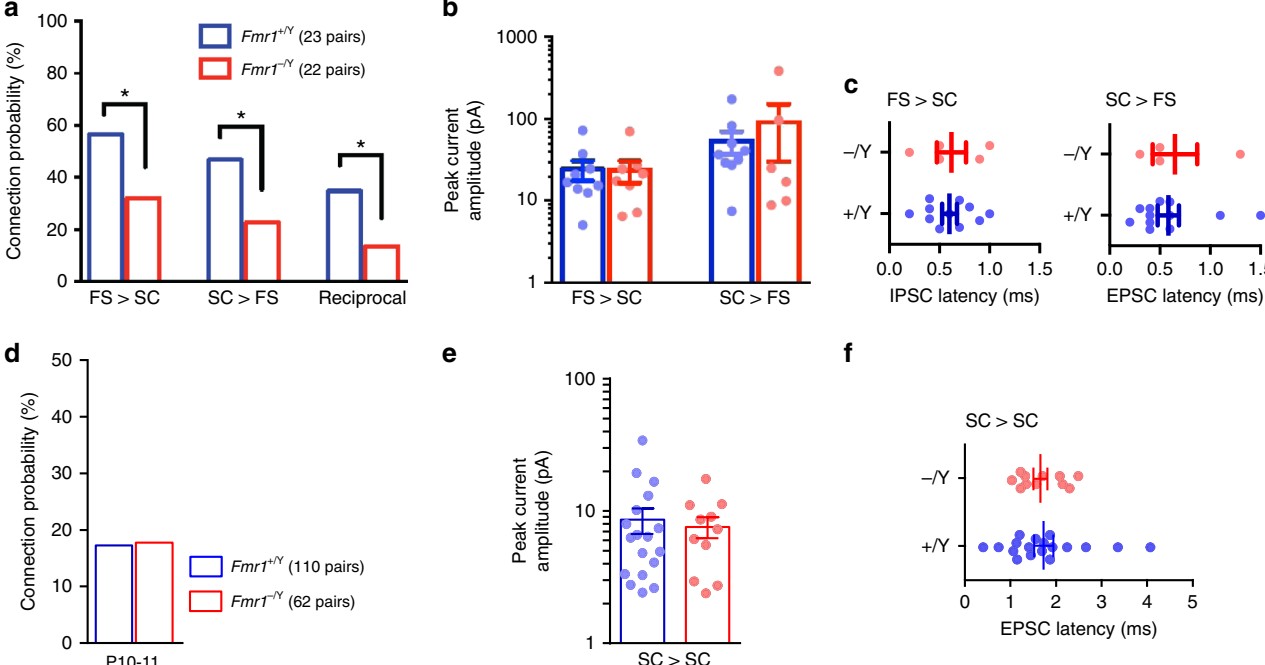

**Fig. 2** Reduced FS IN–SC connectivity in *Fmr1-KO*, but no change in connection properties. **a** Monosynaptic connection probability between pairs of FS and SCs tested with paired whole-cell recording. Asterisks: $p < 0.05$, Chi-squared test. **b**, **c** Connection strengths (**b**), peak evoked monosynaptic current amplitude and current onset latencies (**c**) for connected pairs shown in (**a**). No differences in connection strength between genotype were observed for either connection direction ($p > 0.05$, Mann–Whitney, *N*'s as in (**a**)). **d** Monosynaptic connection probability between pairs of SCs located within the same barrel for recordings in slices taken from P10–11 mice of each genotype. No change in connection probability was observed ($p > 0.05$, Chi-squared test). Not shown: Significant reduction in SC–SC connectivity for *Fmr1−/Y* at P12–15 (37/110, 8/54 tested pairs connected for *Fmr1+/Y* and *Fmr1−/Y*, respectively, $\chi2(1) = 7.78$, $p < 0.01$). **e**, **f** No change in synaptic strengths (**e**) or peak monosynaptic EPSC amplitude (**f**) between the connected SC pairs from P10–11 *Fmr1+/Y* and *Fmr1−/Y* littermate mice shown in (**d**) ($p > 0.05$, Mann–Whitney, *N*'s as in (**a**))

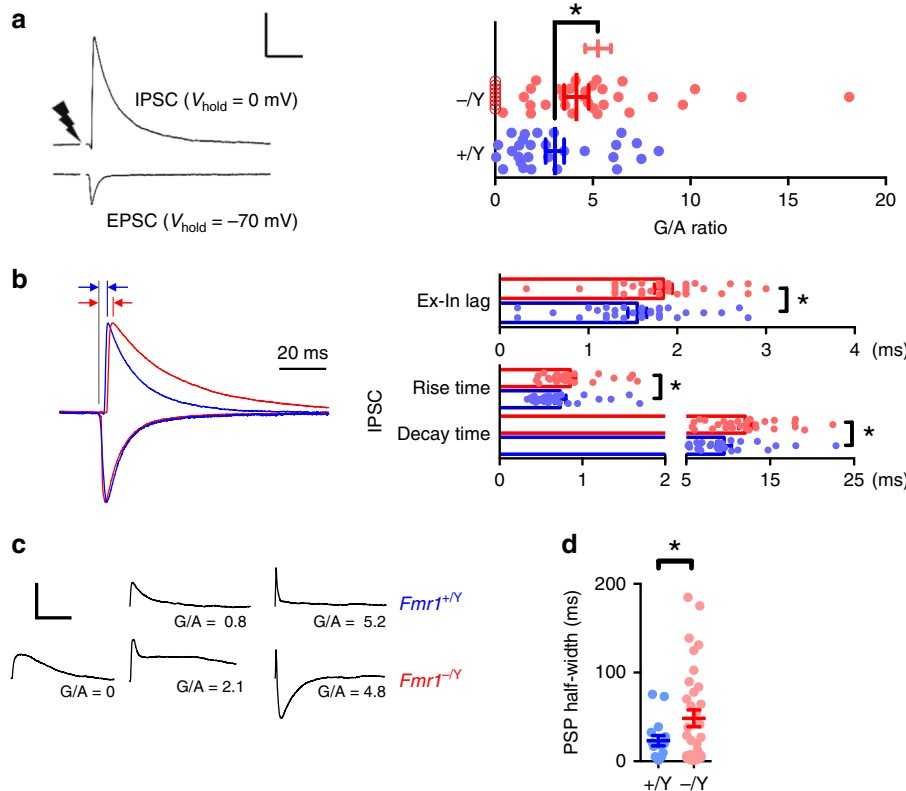

**Fig. 3** Altered thalamocortical FFI in P10–11 *Fmr1-KO* mouse. **a** Left: Trial-averaged voltage-clamp recordings showing direct thalamocortical EPSCs and FF-IPSCs from an example *Fmr1+/Y* Layer 4 SC, Scale: 25 ms/500 pA. The strength of TC-FFI ("G/A ratio") is quantified as the ratio of peak evoked current. Right: Strength of TC-FFI at P10–11. Points indicate neurons, bars are mean ± SEM. Unlike for *Fmr1+/Y* recordings (28 neurons, max. 3 per animal), in *Fmr1−/Y* neurons, some cells (8/38 neurons, max. 3 per animal) lacked FFI; light red bars indicate FFI strength of cells with G/A > 0, hollow markers. Including *Fmr1−/Y* neurons with G/A = 0, average strength of FFI was not significantly different to that of *Fmr1+/Y*, but excluding these neurons (dark red bars and solid markers), the average strength was elevated over that of FFI in wild-type recordings (*Fmr1+/Y* vs. all *Fmr1−/Y* neurons: p = 0.40, *Fmr1+/Y* vs. *Fmr1−/Y* neurons with G/A > 0: p = 0.005, Mann–Whitney). **b** Synaptic kinetics of currents underlying TC-FFI. Left: Example TC-evoked currents in from *Fmr1+/Y* (blue) and *Fmr1−/Y* (red), EPSCs and FF-IPSCs, individually scaled to peak amplitudes. Note the slower decay time constant and onset latency for *Fmr1−/Y* FFI-PSCs (indicated by red and blue arrows). Right: Slower FFI synaptic kinetics for *Fmr1−/Y* SCs. No significant genotype-dependent differences were observed in the same comparisons for kinetics of EPSCs. Asterisks indicate p < 0.05 (t-test), N's (neurons): 23 *Fmr1+/Y*/26 *Fmr1−/Y* (EPSCs); 27 *Fmr1+/Y*/28 *Fmr1−/Y* (FF-IPSCs). **c** Example TC EPSPs from SCs receiving low, medium and high TC-FFI. Note the progressive curtailment of EPSP duration with increasing FFI strength, the lack of a *Fmr1+/Y* example for G/A = 0, and the exaggerated IPSP for the high strength FFI *Fmr1−/Y* example. **d** Slower TC EPSPs in *Fmr1−/Y*. Full width at half-height ('half-width') of EPSPs from SCs (N (neurons) = 19 *Fmr1+/Y*, 36 *Fmr1−/Y*), bars are mean ± SEM, asterisks denote p < 0.05 (t-test). Not shown: the dependence of *Fmr1-KO* thalamocortical EPSP duration on the strength of FFI is distorted. *Fmr1-KO* Neurons with weaker/no FFI (G/A ratio < 2) showed specifically broadened EPSP duration (22.4 ± 7.30 ms vs. 68.4 ± 16 ms, *Fmr1+/Y* vs. *Fmr1−/Y*, p = 0.02), whereas this effect was dampened in neurons with stronger FFI (13.2 ± 8.70 ms vs. 37.1 ± 15.8 ms, *Fmr1+/Y* vs. *Fmr1−/Y*, p = 0.07)

synaptic feed-forward IPSC in the same SCs (at 0 mV, following methods in ref. [32]). Here, we report the E/I balance as the strength of evoked inhibition normalised to the driving excitatory input strength: In SCs from *Fmr1-KO* mice, we found an increase in the peak amplitude of the feed-forward IPSC relative to the monosynaptic EPSC ('GABA/AMPA ratio' [G/A]; Fig. 3a). Notably however, a subset of *Fmr1-KO* neurons lacked any FFI, whereas FFI was observed in all tested WT neurons. The feed-forward IPSC also exhibited slower kinetics and a longer onset lag time (compared with the monosynaptic EPSC) in SCs from *Fmr1-KO* mice (Fig. 3b).

To investigate the consequences of these changes in the relative magnitude and timing of FFI we measured the resulting postsynaptic potential (PSP, Fig. 3c). As we and others have previously shown[32,48,59], the postsynaptic potential (PSP) half-width in SCs is strongly influenced by FFI, which truncates the PSP. We found that PSP full width at half-maximum amplitude (half-width) is increased in the *Fmr1-KO* mice (Fig. 3d) suggesting a decrease in functional FFI, despite the increased feed-forward

inhibitory synaptic input onto SCs. This reduction in functional FFI is likely due to the changes in passive membrane properties of SCs; the mechanism is further explored as described later. One additional mechanism that could contribute to the lack of increase in functional FFI in SCs is a change in chloride reversal potential in the *Fmr1-KO* mice. To test this, we used perforated patch-clamp recordings and found no difference in chloride reversal potential in SCs from the two genotypes (Fig. S4).

These findings, therefore, show that there is diminished functional FFI onto SCs in *Fmr1-KO* mice despite an increase in G/A ratio to a single stimulus. Considering the critical role feed-forward inhibition plays in determining action potential generation and timing in SCs[30,48,60], this deficit is likely to have important consequences for layer 4 network function at this crucial stage of activity-dependent development.

**Enhanced short-term synaptic depression in *Fmr1-KO* mice.** Short-term synaptic plasticity is an important mechanism for information processing in cortical networks[61,62] including

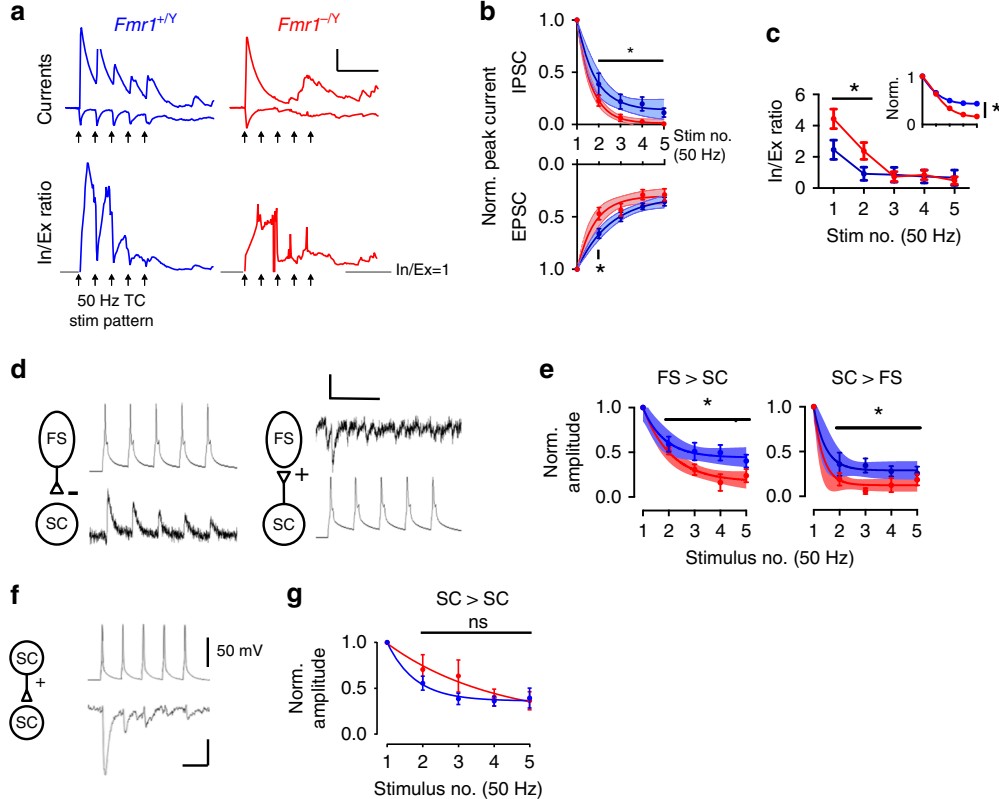

**Fig. 4** Synapse-specific changes to short-term plasticity in P10–11 *Fmr1-KO* mouse. **a** Top: Example voltage-clamped synaptic currents during repetitive TC stimulation at 50 Hz. Note strong amplitude attenuation of TC-evoked and FFI currents in *Fmr1−/Y*. Bottom: Instantaneous G/A ratios for the above traces calculated by dividing FF-IPSC amplitudes by EPSCs for each sampled point in time. Note: (1) the graded attenuation of G/A ratio in the *Fmr1+/Y* and slow onset of G/A balance during stimulus train despite large amplitude FF-IPSC, (2) Temporally disorganised ratio in the *Fmr1−/Y*. Arrows show stimulation times. Scale: 50 ms/100 pA, G/A = 1. **b** Short-term depression of EPSCs and FF-IPSCs during 5 × 50 Hz stimulation: Evoked current amplitudes normalized to steady-state amplitude. Error bars: mean ± SEM normalized peak amplitude after for each stimulus for *Fmr1+/Y* (blue, N = 19 (EPSCs) and N = 11 (FF-IPSCs) neurons) and *Fmr1−/Y* (red, N = 15 (EPSCs) and N = 12 (FF-IPSCs) neurons). Asterisks: significantly different stimulus responses between genotypes (t-test, $p < 0.05$). Shaded regions are best ± 95% CI fits to bi-exponential decay functions. For both EPSCs and FF-IPSCs, the rate of depression for *Fmr1−/Y* responses was faster and a single (i.e. common) fit could not adequately explain the behaviour of both genotypes (Extra sum-of-squares F-test, EPSCs: $p = 0.0007$, $F_{(2,163)} = 7.65$, IPSCs: $p = 0.0002$, $F_{(2,111)} = 9.45$, N's as above). **c** Data from (**b**) represented as stimulus-by-stimulus G/A ratios from FF-IPSCs by EPSCs. Inset shows data normalised to starting G/A ratios. Asterisks indicate stimuli with significant reductions in G/A ratio for *Fmr1−/Y* data (t-test, N's as in (**b**)). **d** Example short-term depression (50 Hz stimulus frequency) of unitary connections between FS-SC and SC-FS neurons (left, right) from paired recordings. Single trials shown. Scale: 50 mV, 10 pA/50 ms. **e** Short-term plasticity analysis in (**b**) but for unitary connections tested between connected pairs of FS and SC neurons (Asterisks: $p < 0.05$, t-test, fits: ($p < 0.05$, Extra sum-of-squares F-test N: FS to SC *Fmr1+/Y* = 9, *Fmr1−/Y* = 8, SC to FS: *Fmr1+/Y* = 9, *Fmr1−/Y* = 6). N's indicate neurons. **f** As for (**d**) but for example connected SC–SC paired recording. Scale: 50 mV, 5 pA/25 ms. **g** Short-term depression of SC–SC connections was indistinguishable between genotypes ($p < 0.05$, t-test, N = 17 *Fmr1+/Y*, 11 *Fmr1−/Y*)

determining the efficacy of FFI[30–32,48]. We therefore compared short-term depression of the TC excitatory input and of the feed-forward TC-evoked IPSC onto SCs. Both excitatory and feed-forward inhibitory transmission in the *Fmr1-KO* mice showed an increase in short-term depression during trains of five stimuli over a frequency range from 5 to 50 Hz compared with WT (Fig. 4a). However, the relative increase in short-term depression was greater for the IPSC than the EPSC leading to a stronger increase in E/I ratio during trains of activity in *Fmr1-KO* compared with wild-type mice (Fig. 4b, c). We also investigated short-term plasticity at SC-FS and FS-SC synapses in connected simultaneous recordings. In contrast to the normal short-term plasticity observed by Gibson et al.[24] in *Fmr1-KO* recordings at 2 and 4 weeks old, we found increased short-term depression at P10/11, both for the excitatory input onto FS from SC and for the inhibitory input from FS to SC (Fig. 4d, e). In SC to SC connections, although there was a trend for a decrease in short-term depression during the train, there was no robust change observed (Fig. 4f, g).

**Increased postsynaptic summation and spiking in *Fmr1-KO* mice**. SCs are the output neurons for layer 4 barrel cortex and serve to integrate TC input and transform it to action potential output, modulated by local feed-forward inhibition[46,49,63]. Our results thus far reveal a number of cellular and synaptic phenotypes in layer 4 barrel cortex neurons in the *Fmr1-KO* mice that likely will effect SC integration of TC input. For FFI, although we show that the G/A current ratios are typically elevated in SCs from P10–11 *Fmr1-KO* mice compared with WT, the activity-dependent depression of FF inhibitory input is faster and greater in KO SCs compared with the depression of thalamic EPSCs (group data in Fig. 4c). Coupled with the increased intrinsic excitability of the KO neurons, this effect is predicted to produce an increase in voltage summation and contribute to abnormal AP generation in response to thalamic input.

Therefore, to investigate the net effect of these phenotypes on the layer 4 circuit we studied the membrane potential response in SCs elicited by TC stimulation. Using whole-cell patch-clamp recordings in current clamp, we found that trains of five stimuli at

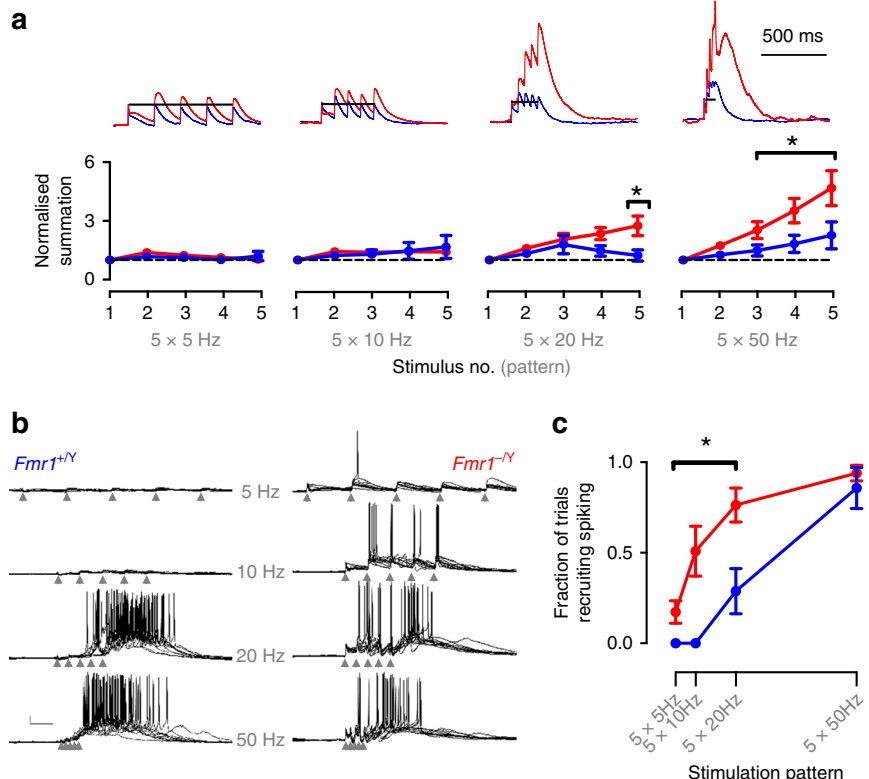

**Fig. 5** Relaxed coincidence detection impairs frequency gating in *Fmr1-KO* L4 networks. **a** Hyper-summation of high-frequency thalamocortical input in *Fmr1*$^{-/Y}$ SCs. Top: example current-clamp recordings showing voltage summation in response to five regular stimuli at frequencies between 5 and 50 Hz. Amplitude is normalized to that of steady-state EPSP. Below: Mean ± SEM normalised EPSP amplitude as a function of stimulus number. Short trains of TC stimuli at 20 and 50 Hz evoked stronger voltage summation in *Fmr1*$^{-/Y}$ recordings: asterisks denote significantly elevated responses ($p < 0.05$, unpaired *t*-test, *Fmr1*$^{+/Y}$ $n = 7$, *Fmr1*$^{-/Y}$ $n = 7$). **b** Shifted sensitivity of L4 network to thalamocortical input frequency in *Fmr1-KOs*. Example current-clamp recordings (10 trials overlaid) showing transient network activity evoked by five repetitive thalamocortical stimuli at 5, 10, 20 and 50 Hz (scale bar: 100 ms, 10 mV). Note relaxed requirement of high-frequency stimulation for generating sustained intracortical activity in *Fmr1*$^{-/Y}$ SCs. **c** Fraction of trials evoking network activity as a function of stimulus pattern. Stimulation frequencies below 20 Hz could not evoke firing (p(spiking) = 0 ± 0) in *Fmr1*$^{+/Y}$ slices, but with low-moderate probability in *Fmr1*$^{-/Y}$ slices. Asterisks denote stimulation frequencies demonstrating significantly elevated firing probabilities in *Fmr1*$^{-/Y}$ recordings ($p < 0.05$, unpaired *t*-test, *Fmr1*$^{+/Y}$ $n = 12$ slices from eight animals, *Fmr1*$^{-/Y}$ $n = 10$ slices from 10 animals). The 5 × 50 Hz stimulation pattern could evoke firing with high reliability for both genotypes (p(spiking): *Fmr1*$^{+/Y}$ = 0.87 ± 0.11, *Fmr1*$^{-/Y}$ = 0.85 ± 0.09). Data represented in b is an extended dataset to that shown in Fig. 9f, g in Booker et al.[64]

frequencies from 5 to 50 Hz produced greater postsynaptic summation in SCs from *Fmr1-KO* mice (Fig. 5a) and a larger fraction of stimuli eliciting action potentials (Fig. 5b, c). We used cell-attached patch recordings to make less invasive measure of spike output. Using this approach, we confirmed that SCs from *Fmr1-KO* mice produce spike output at lower TC stimulus frequencies than wild type. We further found that the timing precision of action potential firing during stimulation (50 Hz), a physiologically relevant frequency, was strongly impaired, exhibiting a lower instantaneous frequency and greater variability in rate and timing (Figs. 6a–d, S5). These compound effects on trial-to-trial fidelity and lower firing rate by *Fmr1-KO* neurons reduced their population average spike density function (Fig. 6b, c individual neurons shown in S5C,D). No significant genotype-dependent changes were observed in the number of spikes fired per trial. Thus, the net effect of the cellular and synaptic changes in the *Fmr1-KO* mice is that SCs that produce action potentials more readily, but with less precise timing.

**Antagonistic cellular and synaptic changes in *Fmr1-KO* mice.**
Our experimental data show that loss of *Fmr1* produces a variety of cellular and synaptic alterations leading to an overall change in the transformation of TC input to action potential output by

SCs. The various mechanisms we describe likely interact in a complex manner to produce the overall circuit phenotype. To better understand this, we first developed a single-cell model of thalamocortical integration to systematically explore the inter-action of our findings in *Fmr1-KO* SCs. We used a leaky integrate-and-fire model neuron with a single dendritic seg-ment, receiving bulk glutamatergic synaptic input connected to a soma receiving somatic GABAergic input, representing tha-lamocortical and FFI inputs, respectively. We adjusted the intrinsic and synaptic kinetics and short-term plasticity beha-viour of the model to match WT and *Fmr1-KO* mean values and altered the strength and timing of the GABAergic input (relative to that of the glutamatergic) to explore the parameter space described by our experimental data (Fig. 7a). We asked which frequencies of train stimulation between 5 and 50 Hz led to spike firing by the model (Fig. 7b). In the absence of FFI (G/A = 0, GABAergic conductance silenced) both WT and *Fmr1-KO* models fired spike(s) at all tested frequencies. Gradual addition of FFI progressively restricted the ability of trains to elicit spikes to those at higher frequencies, but this effect was less pro-nounced in *Fmr1-KO* model compared with WT. Notably, the range of G/A over which this effect was observed are similar to the G/A ranges we measured experimentally. The *Fmr1-KO*

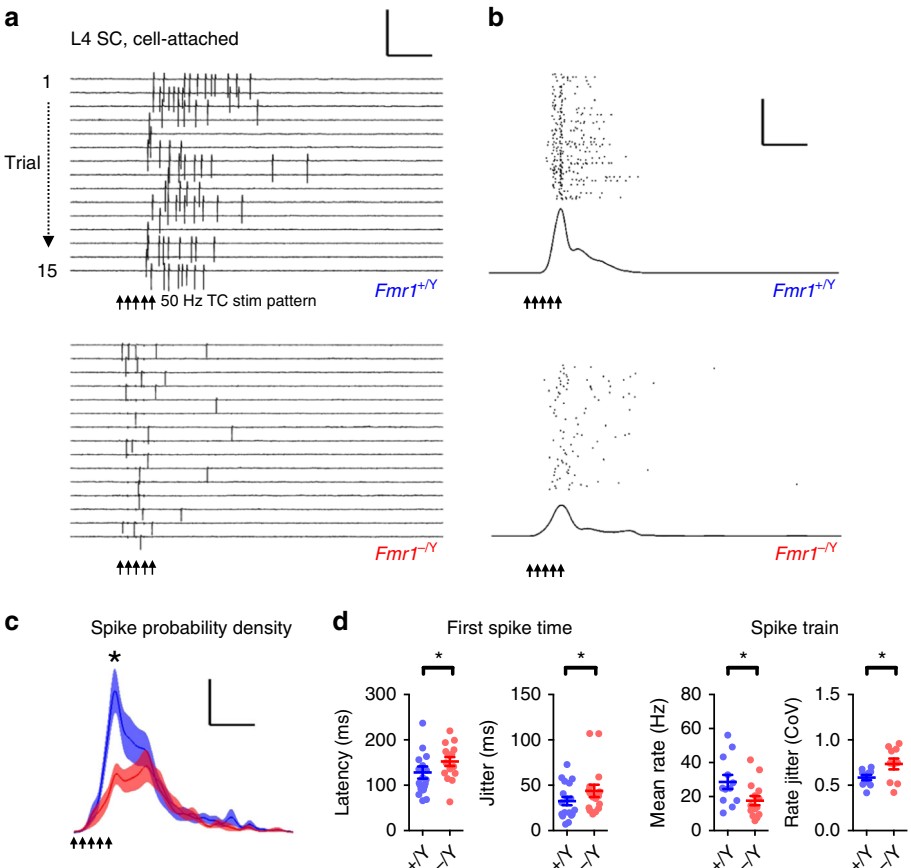

**Fig. 6** Low precision spiking of *Fmr1-KO* Layer 4 excitatory neurons to TC stimulation. **a** Example multi-trial raster of spikes recorded in cell-attached configuration from *Fmr1⁺ᐟ⁺* (top) and *Fmr1⁻ᐟ⁺* (bottom) SCs in response to TC-evoked L4 network activity at 50 Hz. Scale: 200 pA/100 ms. **b** Example calculation of spike density functions for *Fmr1⁺ᐟ⁺* and *Fmr1⁻ᐟ⁺* example neurons. Two hundred consecutive trials showing trial-trial variability in the timing of spikes recorded in cell-attached configuration (different cells from those shown in (**a**)). Bottom: Trial-averaged spike density estimate across trials for neurons shown above. Scale: 50 trials/100 ms. **c** Mean ± SEM spike probability density functions for responding SCs during the peri-stimulus period of TC-evoked network activity. Mean peak spike probability was reduced in *Fmr1⁻ᐟ⁺* recordings, calculated across the whole 1 s post-stimulus sampling period (*Fmr1⁺ᐟ⁺*: 0.023 ± 0.002 sipkes s⁻¹, vs. Fmr1⁻ᐟ⁺: 0.015 ± 0.0016 spikes s⁻¹, $p = 0.008$, t-test, N: *Fmr1⁺ᐟ⁺*: 21 neurons, *Fmr1⁻ᐟ⁺*: 16 neurons). Mean spike density averaged across the successive 200 ms window of was not significantly different between genotypes (t-test, $p = 0.7$). Scale: p(spike/5 ms) = 0.5%/100 ms. **d** Left: Spike-time statistics for the first spike fired per trial for SCs. Mean latency (left) and inter-trial precision (right) were significantly slower and reduced in *Fmr1⁻ᐟ⁺* neurons ($p = 0.01$ and $p = 0.03$, respectively. t-test, N's as in (**c**)). Right: Spike rate and rate stability was significantly different in *Fmr1⁻ᐟ⁺* recordings compared with *Fmr1⁺ᐟ⁺* littermates: SCs fired at rates that were slower and more variable between trials. Plotted points are individual neurons, bars show mean ± SEM values. Asterisks indicate statistically significant differences between genotypes ($p < 0.05$, t-test, N's as in (**c**))

model also reproduced the experimentally observed increases in first spike delay, spike timing jitter and spike number compared with WT (Fig. 7b insert, c).

We used our single-cell model to explore the role of groups of parameters in regulating spike output generated by thalamocortical input over a range of G/A values (Fig. 8). To simplify the model parameter landscape to those involving similar physiological processes, we assigned parameters into four groups. These were namely: short-term plasticity (coefficients constraining synaptic depression behaviour of TC and FFI inputs during repetitive stimulation), excitatory-inhibitory synaptic input delay (lag between simulated TC and FFI input onset times), intrinsic excitability (leak conductances and capacitance of model), and synaptic kinetics (rise and decay kinetics of synaptic conductance). We then systematically replaced *Fmr1-KO* values with WT values for the various parameter groups to simulate different rescue scenarios and measured spike output over a range of G/A values for all of these combinations (four parameter groups to the power of two potential genotypes = 16 simulated scenarios) in response to 5 stimuli at 5–50 Hz. For example, when we 'rescued' excitatory-

inhibitory synaptic input delay on the *Fmr1-KO* background there was an 86% increase in spike output by the model across the parameter space explored, relative to the number of spikes fired by the WT model. Conversely, rescuing intrinsic excitability on the *Fmr1-KO* background produced a 26% reduction in spike output. Thus, manipulation of different parameter groups can have opposing effects on spike output. Overall, by rescuing various combinations of parameter groups we observed that some have antagonistic effects on spike output (e.g. simultaneous rescue of intrinsic excitability and excitatory-inhibitory synaptic input delay) while others drive spike output in the same direction (e.g. excitatory-inhibitory synaptic input delay and short-term plasticity). This analysis indicates that the elevated intrinsic excitability observed in *Fmr1-KO* SCs is a dominating feature driving the changes in spike output. However, changes to other parameters limit the physiological manifestation of this cellular phenotype at the output level of spike generation, suggesting that some of the parameter changes observed in the *Fmr1-KO* thalamocortical phenotype reflect a compensation in response to the underlying pathology that limits circuit dysfunction.

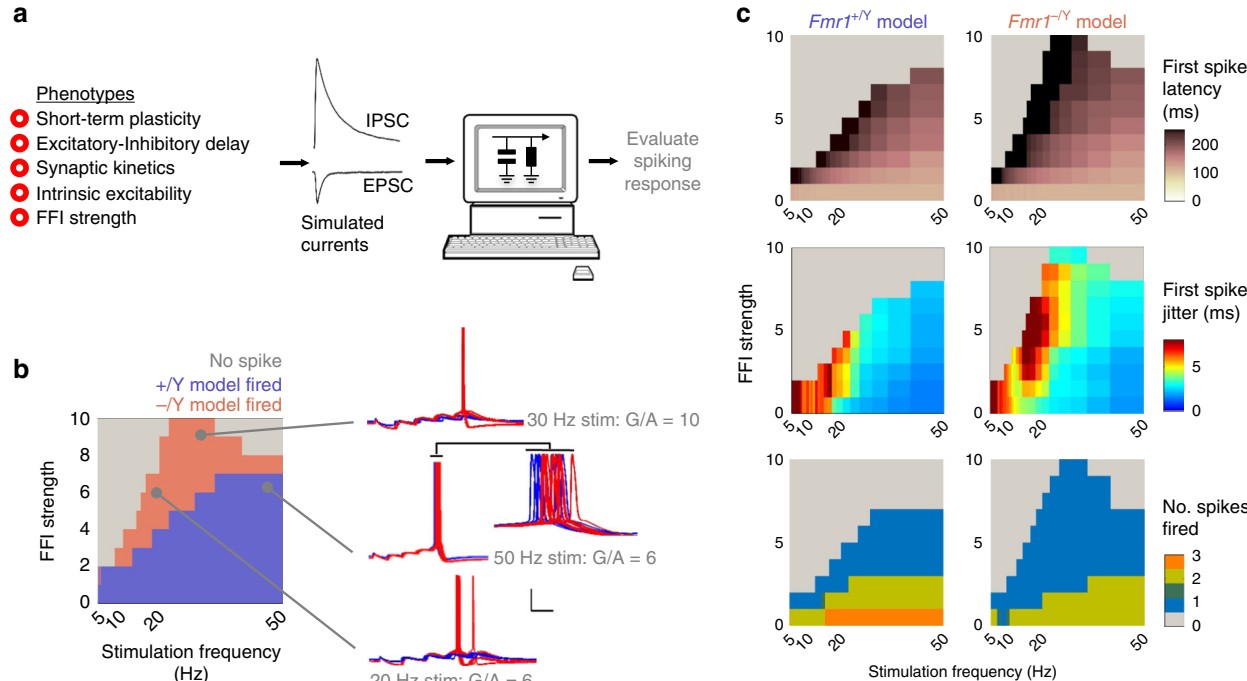

**Fig. 7** Modelling TC summation recapitulates spiking phenotypes of *Fmr1-KO* layer 4 SCs. **a** Schematic of modelling approach. Left: Five grouped covariates measured from *Fmr1+/Y* and *Fmr1−/Y* recordings used in simulation. Centre: Simulated synaptic inputs were tuned with kinetics of recorded currents. Right: Parameter spaces were explored in silico to find conditions that either enhanced or suppressed firing in the *Fmr1-KO* model compared with the WT model. **b** Left: Input frequency dependence of simulated spiking responses for model Layer 4 neurons receiving different strengths of FFI (G/A ratios between 0 and 10 tested). Coloured areas for each modelled genotype indicate combinations of FFI strength/stimulation frequency at which the models fired at least one spike per trial. Red indicates firing parameter ranges in addition to those of the WT model. Note: (1) moderate strength FFI in the WT model prevents spike firing even at high input frequencies (2) the increased number of simulation conditions leading to spiking in the *Fmr1−/Y* model, (3) the insensitivity of spiking regulation in the *Fmr1−/Y* model to inhibitory tone even with FFI strengths elevated to extreme levels (10 trials overlaid). Right: example traces for simulated spiking by the two models at different parameter combinations. Inset: note later and more variable spike times in the *Fmr1−/Y* model. Scale: 20 mV/50 ms. **c** In addition to affecting the overall spike firing response shown in (**b**), genotype-dependent effects were observed in the latency, timing variability and count of spikes fired. Spikes fired later and with lower temporally precision in the *Fmr1−/Y* model across a broad range of model conditions, even with the FFI strength increased to the extreme values as observed in the *Fmr1−/Y* recordings. More conditions led to spiking in the *Fmr1−/Y*, despite a slight decrease in numbers of spikes fired per trial across the distribution compared with *Fmr1+/Y* simulations

**Network dysfunction in layer 4 in *Fmr1-KO* mice.** Having explored the mechanisms underpinning the altered transformation of the thalamocortical input in layer 4 *Fmr1-KO*, we next asked how the cellular and synaptic deficits result in altered network activity and what the consequences are for information processing by layer 4. We developed a spiking network model of a cortical layer 4 barrel, based on our electrophysiological recordings (Fig. 9a and S6; see also Methods and Supplemental Table 1). The model consists of a network of interconnected SCs and FS, stimulated by TC input. The great advantage of layer 4 barrel cortex in this regard is the numbers of SCs and FS in a barrel are known as is their probabilistic connectivity[43]. The model was randomly connected, constrained by the experimentally determined probabilities (as detailed above) for WT and *Fmr1-KO* phenotypes. Cells were represented as single-compartment leaky integrate-and-fire (LIF) neurons connected by inhibitory (GABA$_A$) and excitatory (AMPA and NMDA) synapses displaying short-term plasticity. In addition, model parameter distributions matched recorded biological diversity in intrinsic neuronal properties, synaptic strength and kinetics, transduction delays and short-term plasticity of synapses (Fig. S6). Simulated membrane depolarisation and action potentials were measured in response to the same patterns of train stimulation of TC axons as we used in the slice experiments.

Ab initio, the random model reproduced well the main features of our experimental data in both WT and *Fmr1-KOs*, both at the single cell and at the network level. The *Fmr1-KO* displayed increased subthreshold summation and increased spiking activity to TC input at low train frequencies at the single-cell level (Fig. 9b, c). Similar to our experimental findings and in agreement with our single-neuron thalamocortical model, the timing of first spikes fired in the *Fmr1-KO* model was delayed and showed decreased inter-trial fidelity (Fig. S7a). Similarly, fluctuations in trial-to-trial mean firing rate were enhanced in the *Fmr1-KO* model (Fig. S7b). Notably, approximately half of the per-trial-averaged numbers of spikes fired by neurons in the recurrently connected model could be accounted for by those fired by our single-cell model lacking recurrent connectivity (Fig. 7c cf. Fig. S6a) demonstrating a strong thalamocortical contribution to the generation of spiking in layer 4.

At the network level, high-frequency TC stimulation evoked a transient reverberating population response, characterised by high rates of simultaneous model neuron firing (highlighted by peaks in the grey firing histograms overlaid in Fig. 9d). In the WT model these synchronous population events were initially tightly locked to latter stimuli during the train before transitioning to self-sustaining network activity persisting after the last stimulus. *Fmr1-KO* model network activation required fewer stimuli and was recruited immediately following the first TC-locked spike volley but did not produce the self-sustaining network activity observed in WT. We explored this observation further by examining the stimulus-evoked population firing rates of excitatory and

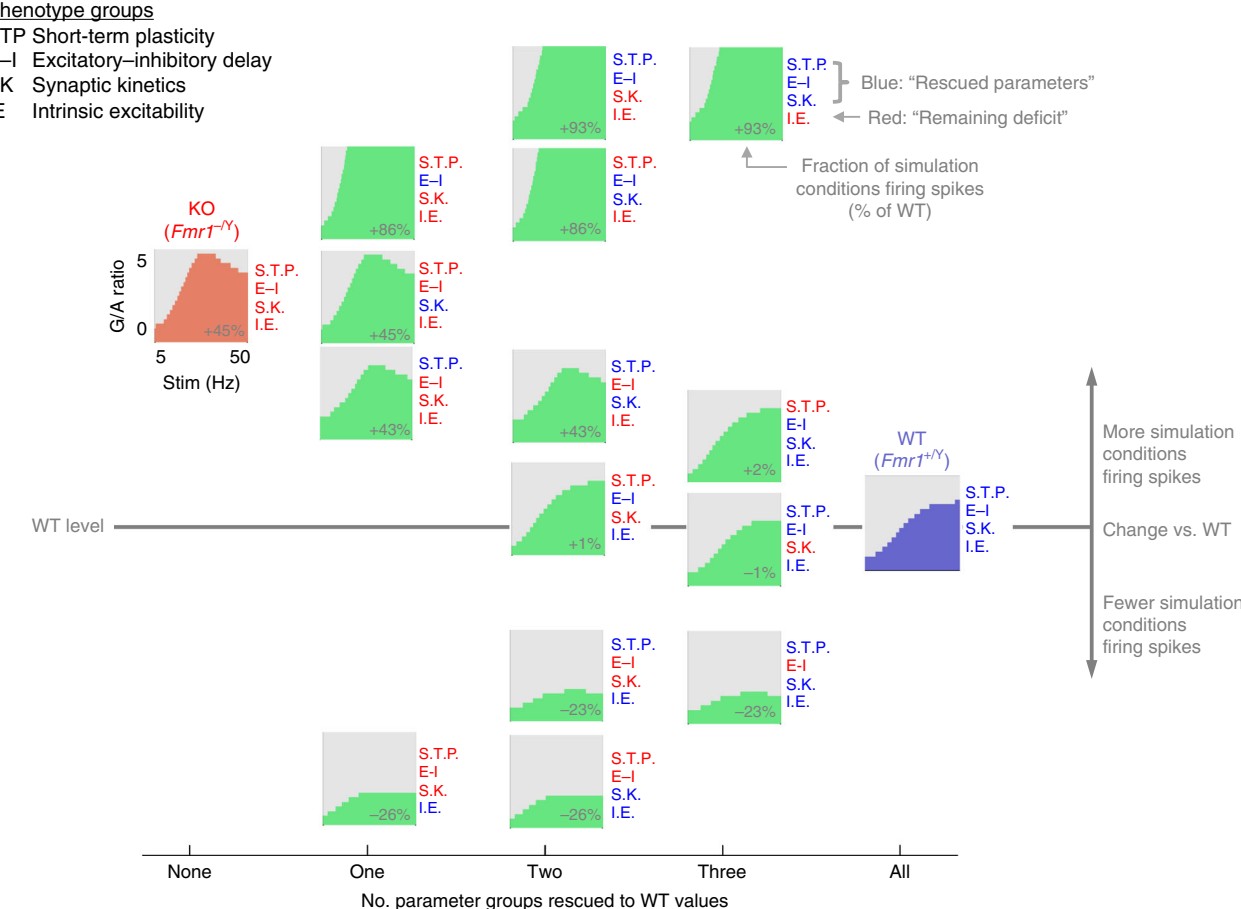

**Fig. 8** Relative contributions of different mechanisms to dysfunction in the *Fmr1-KO* model. Model parameter space explored for 16 different possible combinations of simulated *Fmr1+/Y* ('WT') and *Fmr1−/Y* (KO) conditions (four parameter groups, two possible genotypes, i.e. $4^2$ combinations). Spike firing conditions (in response to five repetitive stimuli) are shown in green for each intermediate rescue scenario as well as *Fmr1−/Y* to *Fmr1+/Y* models. For each partial rescue scenario (either with *Fmr1+/Y* or *Fmr1−/Y* values, shown in blue and red, respectively), the total count of spike firing conditions across the whole 5–50 Hz and 0 < G/A < 10 range is shown compared with that of the wild-type simulation (i.e. full *Fmr1−/Y* model fired > 1 spike(s) in 45% more simulated FFI strength/input frequency conditions compared with the *Fmr1+/Y* model)

inhibitory neurons in the model network (Fig. 9e). The first TC stimulus led to spiking in a greater fraction of *Fmr1-KO* inhibitory neurons compared with WT. Strong reductions in the fraction of firing of inhibitory neurons were observed for successive stimuli for both genotypes, but this was much greater for the *Fmr1-KO* model, consistent with experimental data. In the WT model, firing rate increases in the excitatory and inhibitory populations were strong and temporally matched, leading to a slowly decaying limit cycle (Fig. 9f) for recurrent network firing. In contrast, the rates of synchronous firing for excitatory and inhibitory populations were disorganised in the *Fmr1-KO* model, with lower overall synchrony, producing unstable limit cycles.

Overall, the network model captures many of the important features of our experimental data and suggests a mechanism by which single-cell physiological defects in the *Fmr1-KO* layer 4 network (particularly weaker Ex–In connectivity and cellular hyperexcitability) generate impaired network activity.

**Impaired pattern discrimination in a *Fmr1-KO* layer 4 model.** Thus far we have demonstrated that a simple data-driven network model of *Fmr1-KO* layer 4 reproduces the main experimentally obtained differences in responses to patterned TC input. Since a major role of barrel cortex layer 4 is the representation of sensory details in the rate and timing of excitatory neurons action

potentials[65], we next examined how individual neurons and small cell ensembles in our WT and *Fmr1-KO* layer 4 models performed when challenged to detect subtle temporal features of simulated sensory input (Fig. 10). This was represented as an extra stimulus inserted into the input stimulus train and can be conceptualised as the perturbation of TC neurons' firing patterns in response to a small surface bump encountered during active sensation.

The majority of WT Ex neurons detected the oddball stimulus as evidenced by a change in firing rate (indicated by the off-diagonal clustering of points in Fig. 10a). However, the majority of simulated *Fmr1-KO* neurons were less likely to detect the oddball stimulus. In addition, WT neurons represented the presence of an oddball stimulus in the timing of the first spike fired (Fig. 10b) such that, despite an interdependence, WT neurons could encode the presence of an oddball as a change in spike rate and/or a shift in first spike timing. Conversely, *Fmr1-KO* neurons showed a reduced change in first spike timing in response to the oddball stimulus. To quantify both of these spike train effects, we used a distance metric that encompasses both spike train and rate[66,67] to demonstrate the distribution of single-cell oddball stimulus representations (Fig. 10c). The distribution of regular-oddball spike train dissimilarities was uniformly reduced in the KO, implying that the net effect of spike train rate and timing leads to a global reduction across the network in

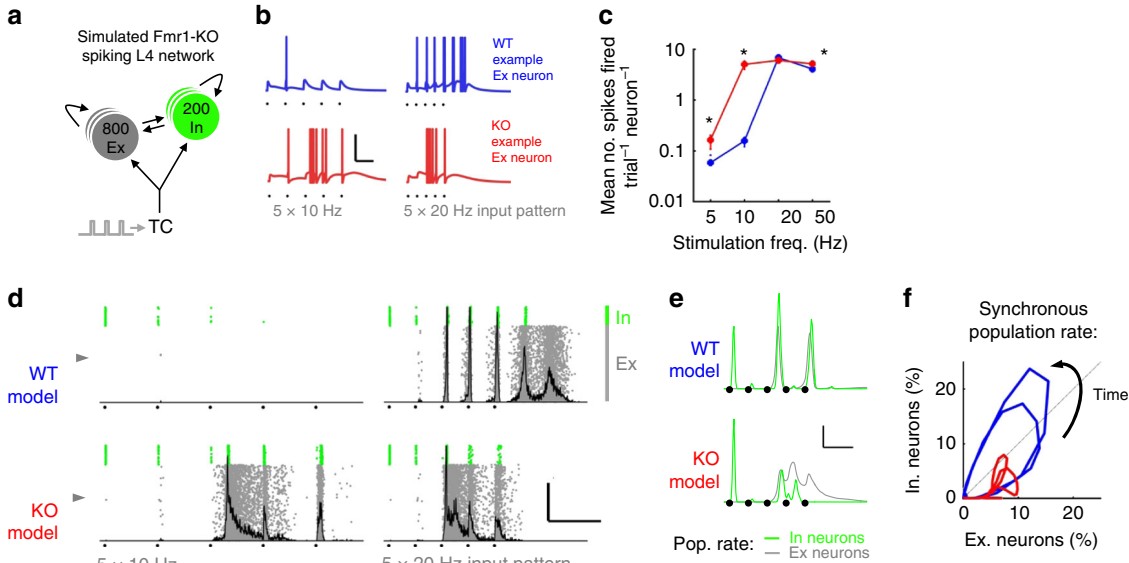

**Fig. 9** A *Fmr1-KO* layer 4 model reproduces features of the network response to TC input. **a** Schematic of model circuit comprising recurrently connected pools of Ex and In neurons receiving simulated thalamocortical input. **b** Simulated membrane potential for two example Ex neurons (indicated by grey arrowheads in (**d**)) for TC input at 10 and 20 Hz stimulation frequency. Black dots indicate stimulus times. Scale: 100 ms/50 mV. **c** Frequency-dependent spike output of model Ex neurons. Asterisks denote significantly different spike mean spike counts between models ($p < 0.05$, *t*-test, 800 neurons per model, 5 random models, average of 10 trials each). **d** Exemplar full network spike rasters for Ex and In neurons (grey and green, respectively) showing firing patterns in response to 5 × 10–20 Hz model thalamocortical stimuli. Population histograms of Ex neurons overlaid in grey. Scale: 100 ms, 50% Synchrony. **e** Grand mean Ex (grey) and In (green) population spike density functions (5 random seeds, 10 repeats each) from $Fmr1^{+/Y}$ and $Fmr1^{-/Y}$ simulations. Note impaired E–I population interaction in *Fmr1-KO* simulations. **f** Phase plot summarizing rhythmic Ex–In population interaction in $Fmr1^{+/Y}$ and $Fmr1^{-/Y}$ models. Note reduced global synchrony and impaired recruitment of Inhibitory neurons in $Fmr1^{-/Y}$ model

the ability to discriminate a fine temporal feature of simulated sensory input.

The principal output from the layer 4 circuit is a sparse population code provided by the ascending synaptic connection to layer 2/3 neurons[51,52,68–71], the activity of which plays an instructive role in the development of the layer 2/3 network[44,72]. Therefore, it is important to consider how the defects in layer 4 stimulus representation might relay at the subsequent processing level. We therefore used a linear population decoder[73] to classify input stimulus patterns based on the layer 4 output firing pattern (Fig. 10d). For population codes formed from ensembles of between 10 and 500 neurons (i.e. representing convergence of approximately 1/80 to 2/3 of model layer 4 neurons to a layer 2/3 'readout' neuron), more neurons were required for the *Fmr1-KO* model to reach the same degree of classification accuracy as the WT model (Fig. 10e). Even drawing 500 neurons (~2/3 of the total excitatory network), population coding of input pattern was impaired in the *Fmr1-KO* model.

Taken together, this result suggests that the developing synaptic pathway from layer 4 to layer 2/3, as schematised here as the layer 4 network's input pattern classification accuracy would be mis-informative about the quality of sensory input in the *Fmr1-KO* model, and that information gleaned from a larger pool of layer 4 neurons would be necessary for an informative representation of the sensory input.

## Discussion

Altered sensory reactivities are a common and debilitating feature of FXS and of ASDs more generally (reviewed in refs. [1,74]). Sensory symptoms are often distressing for affected individuals as they lead to anxiety and social withdrawal, as well as self-injurious behaviour. Unfortunately, very little is known about when and how altered sensory function arises during development, nor

how altered neuronal properties affect experience-dependent development.

Here, we dissect the cellular mechanisms by which FMRP deletion results in a paradoxical increase in both G/A ratio and feed-forward summation of TC input induced layer 4 SC EPSCs (see Supplementary Discussion for comparison to existing literature on altered FFI in *Fmr1-KOs*). Despite this increase in local excitability in the layer 4 feed-forward circuit, stimulation of TCs at behaviourally-relevant frequencies leads to decreased and disorganised spiking of L4 SCs during recurrent activity. Modelling this range of phenotypes predicts an altered temporal coincidence detection of thalamocortical inputs at a range of physiologically relevant stimulation frequencies, a prediction that is borne out by experimental data. Together these data indicate that the sensory information being relayed to layer 2/3 is dramatically altered at the commencement of whisking (P12), which would have a deleterious effect on experience-dependent development and hence sensory processing at later ages. These findings also highlight the power of integrating experimental and mathematical approaches to understanding and forming testable hypothesis about the nature of the developmental mechanisms underlying neuronal circuit dysfunction in ASD.

An altered E/I balance has been causally linked to a range of psychiatric disorders from ASD to schizophrenia[21,75–78]. In primary somatosensory cortex, the E/I balance is essential for regulating both the frequency (spike rate) and quality (spike timing) of action potentials, and hence for accurately processing sensory input[65]. In particular, the precise temporal tuning of excitation and inhibition is critical to signal processing of cortical networks[79], as reported elsewhere[80]. Here, we find a reduction in the E/I balance in a subset of layer 4 SCs that causes an overall decrease in the E/I at the population level despite the observation of a subset of *Fmr1-KO* SCs that completely lack functional FFI. The reduction in the E/I balance is accompanied by a gain in

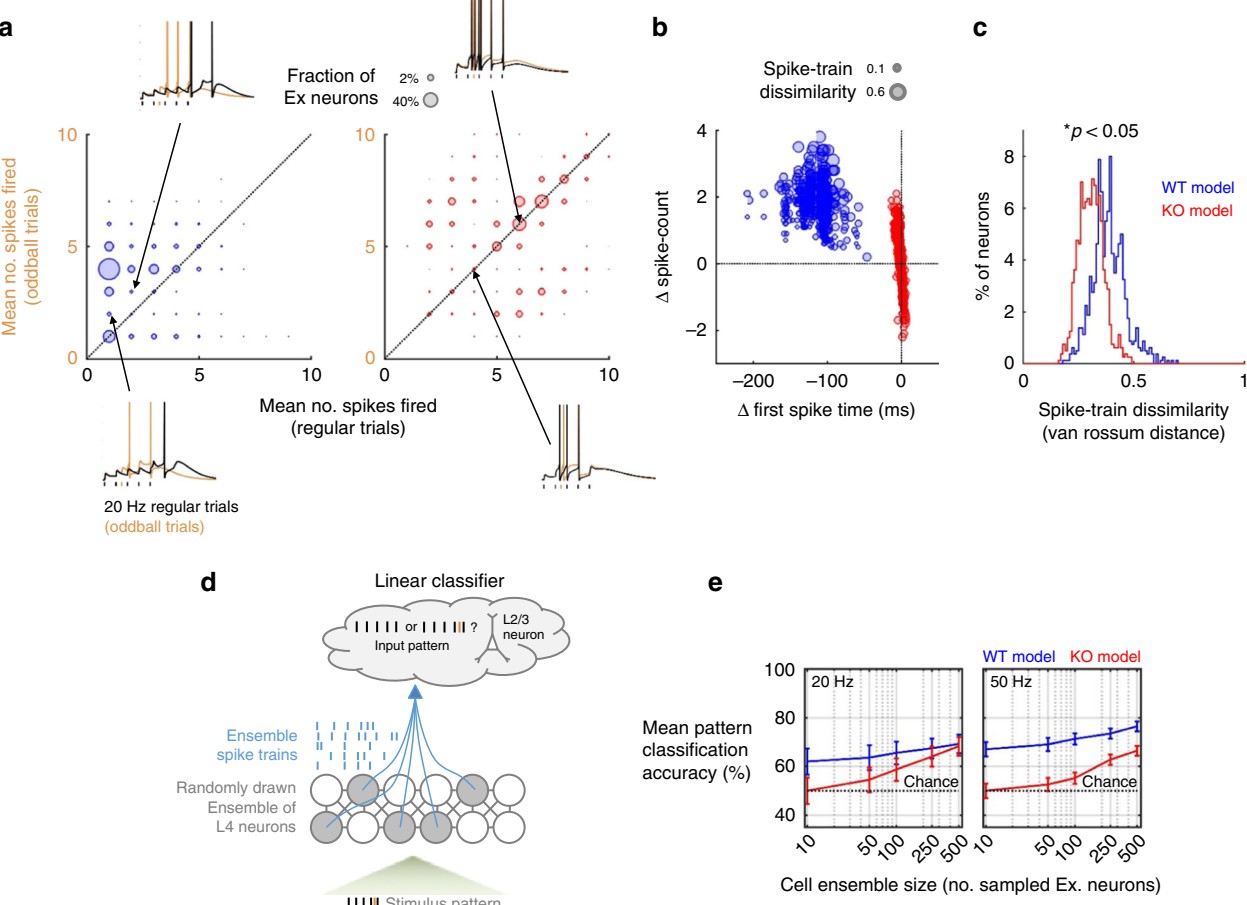

**Fig. 10** Impaired sensory coding by neural ensembles in a model of *Fmr1-KO* layer 4 network. **a** Network-wide representation of an extra oddball stimulus inserted into a regular stimulus train (orange tick, single trial example) by firing changes. Circle sizes: fraction of neurons at each coordinate. Insets: simulated $V_{membrane}$ of representative neurons for regular (black) and oddball (orange) trials. *Fmr1*[+/Y] model neurons typically increased firing rates on the oddball trial, but *Fmr1*[−/Y] neurons only weakly represented the presence of the oddball stimulus by change in firing rate (bulk of points on diagonal). **b** Different coding schemes underlie representation of sensory input details for simulated *Fmr1*[+/Y] and *Fmr1*[−/Y] networks. *Fmr1*[+/Y] neurons typically increased their firing rate and advanced their first spike in response to an extra oddball stimulus. *Fmr1*[−/Y] neurons showed inflexibility in their spike rate but a bidirectional change in first spike time. Size of points denotes mean spike train dissimilarity for each neuron (van Rossum distance between spike trains on regular and oddball trials). **c** Population histogram of spike train encoding of oddball vs. regular input patterns. Population average oddball sensitivity of individual cells was reduced in the *Fmr1*[−/Y] network model (Kolmogorov–Smirnov test). **d** Classification of input pattern from the spike trains of cell ensembles in the layer 4 model, analogous to readout of input to layer 4 by layer 2/3 neurons. Schematic illustration of random sampling of neurons from the layer 4 network model for input to a linear classifier. **e** Impaired coding of input detail by ensembles of SCs neurons in the *Fmr1*[−/Y] model. Mean leave-one-out decoder cross-validation error for *Fmr1*[+/Y] and *Fmr1*[−/Y] networks for randomly drawn ensembles of varying sizes between 10 and 500 neurons. At 20 Hz input frequency, ensembles comprised of 10 or 20 *Fmr1*[−/Y] neurons performed significantly worse than *Fmr1*[+/Y] ensembles (*t*-test, $p < 0.05$), and no better than chance at 20 and 50 Hz (*t*-test vs. responses with permuted stimulus labels, $p > 0.05$). All *Fmr1*[+/Y] ensemble sizes performed better than chance at both frequencies. At 50 Hz, all *Fmr1*[−/Y] ensemble sizes performed worse than corresponding *Fmr1*[+/Y] ensembles (*t*-test, $p < 0.05$)

overall excitability because of elevated SC intrinsic excitability and an increase in the delay between TC excitation and FFI onto spiny stellate neurons (Ex–In lag). This delay results in a broadening of the TC EPSP and a large increase in the summation of EPSCs following TC stimulation that is especially pronounced at higher stimulation frequencies. Compounding this increase in excitatory summation, we find a more dramatic loss of inhibition relative to excitation during physiologically relevant repetitive stimulations. The rapid decrease of inhibition results in a further increase in summation of the EPSPs. These data highlight the vagaries of simply examining E/I imbalance as a key characteristic of circuit dysfunction. It is important to remember that the E/I balance is dynamically regulated[51,79,81–83] and is only one factor that regulates neuronal processing within a complex circuit and its role in mediating sensory dysfunction[9,10]. These nuanced and dynamic circuit effects further highlight the need for

reductive circuit modelling in disentangling the processes underlying circuit dysfunction in animal models of psychiatric disease[80,84].

Gibson et al.[24] have previously reported normal thalamocortical FFI in the *Fmr1-KO* mouse at P14 and P21, implicating impaired excitatory drive onto layer 4 FS neurons as the mechanism underlying layer 4 network hyperexcitability. The disparity between these findings and the present study could result from the age of the animals being examined and the static nature of single-stimulus responses to thalamic stimulation employed by Gibson et al., which lacks the information of the dynamic circuit behaviour invoked by our stimulation paradigms.

Our finding of an increase in input resistance of layer 4 SCs is in good agreement with previous findings for this cell type[24] and is perhaps the most consistent physiological feature across neuronal cell types in the *Fmr1-KO* mouse[24,85–87]. It is important to

note that several *Fmr1-KO* studies spanning prefrontal and entorhinal cortices as well as hippocampus have demonstrated altered spike firing and excitability in the absence of altered input resistance[88–90], indicating that alteration in conductances contributing to cellular input resistance are only one (potentially co-occurring) mechanism underlying distorted cell-intrinsic response to input. A further pertinent observation supporting this claim comes from the MeCP2 mouse model of Rett syndrome[91,92], in which levels of cortical neuronal firing are dampened by reduced E/I tone in the absence of any change to intrinsic cellular excitability. In the present study, elevation in input resistance and resultant slowing in membrane time constant raises the impedance to sinusoidal stimuli. By increasing the intrinsic sensitivity of *Fmr1-KO* SC neurons to low frequency stimulation this has the compound effect of many more action potentials being generated in response to low frequency stimuli, and a dampening of responsivity to high-frequency inputs. Importantly, these input resistance-driven changes in intrinsic properties also result in a phase shift in the voltage response to input currents, for example producing a phase shift in action potential generate. Such a shift in phase can have dramatic effects on spike-time dependent plasticity (STDP[93]) which may contribute to the shift in the timing of the critical period for LTP at thalamocortical synapses[28]. Crucially, the maturation of the layer 4 excitatory network in the brief developmental window beyond P10 is sensitive to sensory experience[33] and relies on spike timing-dependent plasticity mechanisms[93]. The cellular and circuit disruptions to spike-time representation reported here would be expected to dramatically alter experience-dependent development of primary somatosensory cortex in *Fmr1-KO* mice and could contribute to the circuit hyperexcitability reported at older ages. Consistent with this idea, impaired STDP is reported in the *Fmr1-KO* mouse prefrontal cortex[94].

Our findings indicate that the circuit deficits in layer 4 of somatosensory cortex of *Fmr1-KO* mice arise from a range of cellular deficits, however, their relative contribution to the overall layer 4 circuit function and hence the information propagated to layer 2/3 is not known, prompting our use of circuit modelling.

The distributed nature of the physiological changes that result from the loss of FMRP suggest that compensatory changes may be activated even at these early stages of development. However, it is not clear whether the phenotypes reported here, or indeed many of those reported for other cell types (for review see ref. [14]), arise as a direct result of the loss of FMRP or homeostatic compensation in an attempt to restore circuit function. In this context, the recent work of Antoine et al.[84] is particularly relevant. They demonstrated that altered E/I balance in layer 3 of primary somatosensory cortex was a common feature of multiple ASD mouse models. Furthermore, modelling indicated that the changes in E/I balance was tuned to stabilise synaptic drive and spike output for cells near threshold strongly indicating that altered E/I balance was a homeostatic response of the neurons to normalise spiking. For the phenotypes described in this study, it is not possible to determine which cellular features are directly related to the loss of FMRP and which reflect homeostatic changes. However, what is clear is at this early stage of development, E/I balance is not tuned sufficiently to normalise spike output of layer 4 SCs. Future studies examining later stages of development combined with their electrical and/or pharmacological manipulation will be needed to determine the role of each of these physiological parameters in determining the overall circuit dysfunction and the overall hyperexcitability phenotype. It should be noted that, irrespective of causality, each of these phenotypes could provide a potential for therapeutic intervention for FXS but the efficacy of treatment is likely to be specific to particular developmental time-points.

Recent studies have attempted to pharmacologically normalise abnormal E/I balance in mouse models of diverse forms of intellectual disability[92,95–99]. Our findings indicate that E/I balance may be a poor proxy for underlying cellular disturbances and subsequent circuit dysfunction and hence should be approached with caution when devising and determining the effectiveness of potential treatment strategies.

The approach taken in this study was to build a mechanistic understanding of a cellular and circuit dysfunction through combined and bidirectional physiology and predictive modelling. By studying seemingly small pathophysiological changes in the component parts of a well-described dynamic circuit, our models were able to predict large physiological increases in the sensitivity to ethological stimuli. The local circuit model lacks the recurrently connected architecture of L4, thus it is useful to disentangle the contributions of direct vs. feedback processing on the firing patterns of the SCs. The single-cell FFI model specifically predicted the altered summation properties at a range of stimulation frequencies that were verified through experimentation. Furthermore, by substituting each parameter or combination of parameters from *Fmr1-KO* animals with that from their WT counterparts, we were able to begin examining the relative contribution of each phenotype to the overall local circuit dysfunction.

This model also provides a powerful framework to study compensation within neocortical neurons that arise from particular genetic mutations. It could also be used to examine the effects of targeted treatments, for example by simulating the effect of a single parameter repair or combination repair strategy, as we do in the current study.

To encompass the recurrent circuit activity initiated by thalamocortical activity, we generated a network model to investigate the nature of the output of the of the layer 4 circuit. The primary goal of this model was to the understand the quality and fidelity of the information leaving layer 4 in *Fmr1-KO* compared with WT which is key to driving experience-dependent development of layer 2/3 neurons[44]. Using data-tuned cell models as a starting point, we incorporated experimentally derived connection probabilities with known layer SC and FS cell numbers and GABA, NMDA and AMPA currents. The model predicts a decrease in spike firing and loss of trial-to-trial fidelity in the *Fmr1-KO* animal, accurately reflecting our data obtained using patch-clamp recordings from slices. To simulate a subtle temporal sensory disturbance during active whisking, we examined the effect of an oddball stimulus introduced during the stimulus train. The WT model was able to detect the presence of an oddball stimulus through a change in the firing rate and timing of the spikes fired by Excitatory neurons. On the contrary, the *Fmr1-KO* model was less likely to alter its firing rate and the timing of the first spike in response to the oddball stimulus, indicating a robust decrease in the circuit's ability to discriminate changes in the sensory environment. These findings strongly suggest that the ability of the layer 4 circuit to relay details about patterned sensory information, a key feature of experience-dependent plasticity, is dramatically reduced in *Fmr1-KO* mice and make some strong and testable predictions for human psychophysical experiments and their animal experimental in vivo counterparts. The consequence of reduced sensory driven plasticity would be a retardation of circuit development. Importantly, Bureau and colleagues[100] found a delay in the developmental increase in connection probability between layer 4 and layer 2/3 cells in somatosensory cortex of *Fmr1-KO* mice that was mirrored by a delay in the pruning of SC axons projecting to layer 2/3. This delay could be accounted for the shift in the sensitive period for inducing LTP at TC inputs in layer 4 in somatosensory cortex which is delayed in *Fmr1-KO* animals[28]. Importantly while some key features of

somatosensory cortex development are delayed, this is not a global phenomenon. For example, layer 4 SCs show typical developmental trajectories for dendritic elaboration, spinogenesis, and synaptogenesis[28,101]. They also show the typical activity-dependent restriction of their dendrites to the regions of thalamocortical axon terminals[101]. These findings provide support for the idea that a mismatch in the maturation of physiological and/or anatomical properties that are necessary for the subsequent development of circuit function that may underlie pathology in FXS[102].

The mismatch in developmental trajectories demonstrated here may subsequently induce altered development in downstream areas. At P10, we find a dramatic reduction in the information being propagated to layer 2/3 at this age. Hence, for a layer 2/3 neuron, the loss of FMRP essentially mimics sensory deprivation[31–33,93]. Intriguingly, Gainey et al.[103] found a compensatory change in E/I balance following experience-induced plasticity to whisker trimming similar to that found by Antoine et al.[84] in layer 2/3 of older Fmr1-KO mice. Hence the changes in E/I balance seen in layer 2/3 at P21 may result from the altered circuit activity we observe here. Several studies have examined changes in somatosensory circuit function in Fmr1-KO mice at more advanced stages of development[9,24,26,84] and it will be important to determine whether these alteration could similarly arise from the altered circuit activity at younger stages of development and the resulting changes in experience-dependent development.

Despite these differences in age between studies, we have confirmed key findings from previous work. Notably, although we report normal synaptic connection probability between pairs of L4 SCs at P10–11, but this was reduced by P14 in Fmr1-KO mice (reported in legend for Fig. 2d) in agreement with Gibson et al.[24]. We hypothesize that an initial developmental wave of synaptic connectivity emerges unperturbed between SCs in Fmr1-KO L4 despite the delayed critical period for thalamocortical plasticity[28] as well as local circuit and cell-intrinsic effects reported in the present study.

## Methods

**Mice.** Fmr1-KO mice (B6.129-Fmr1$^{tm1Cgr}$, RRID:MGI_MGI:3815018) were obtained from the Contactor laboratory (Northwestern University, Chicago IL, USA). Mice were maintained on a C57Bl6/J background. Hemizygous male Fmr1-KO (Fmr1$^{-/Y}$) and wild-type littermate (WT, Fmr1$^{+/Y}$) pups were used at postnatal days 10–11 (P10–11). All procedures were carried out according to UK Home Office and NIH IACUC guidelines for animal welfare. N's of neurons/animals for each experiment are indicated in the individual figure legends.

**Brain slice preparation.** Mice were decapitated and brains were rapidly removed and placed in ice-cold carbonated (95% $O_2$/5% $CO_2$) cutting solution. Thalamocortical brain slices (400-μm thick) were prepared according to reference[41] using a vibrating microslicer (Leica VT1200). Slices were transferred to artificial cerebrospinal fluid (aCSF) and stored at room temperature. aCSF contained (in mM): NaCl 119, KCl 2.5, NaH$_2$PO$_4$ 1, NaHCO$_3$ 26.5, Glucose 11, MgSO$_4$ 1.3, CaCl$_2$ 2.5. Cutting solution was identical except for substitution with 7 mM MgSO$_4$ and 0.5 mM CaCl$_2$.

**Electrophysiology.** Patch-clamp recordings were performed using internal solutions containing (in millimolar): CsMeSO$_4$ 130, NaCl 8.5, HEPES 10, EGTA 0.5, Mg-ATP 4, Na-GTP 0.3, QX-314 (Tocris) 5 (for voltage-clamp recordings) or KMeSO$_4$ 130, NaCl 8.5, HEPES 5, EGTA 0.5, Mg-ATP 4, Na-GTP 0.3 (for current-clamp recordings) pH 7.4 at 32 ± 0.5 °C, adjusted to 285 mOsm. Junction potential was uncorrected. Slices were superfused with aCSF saturated with 95% $O_2$/5% $CO_2$ at 32 ± 0.5 °C at 8 ml/min, containing in (mM): NaCl 119, KCl 2.5, NaH$_2$PO$_4$ 1, NaHCO$_3$ 26.2, Glucose 11, MgSO$_4$ 1.3, CaCl$_2$ 2.5, adjusted to 305 mOsm, pH 7.4 at 32 ± 0.5 °C. Patch recordings were performed with Multiclamp 700B amplifiers (Molecular Devices), using pipettes (4–7 MΩ open bath resistance) fabricated from thin-walled borosilicate glass (Warner Instruments). Signals were low-pass Bessel filtered at 10 KHz and digitized at 20 KHz using either a Molecular Devices Digidata 1322A board and pClamp 10.2 for recordings performed in the USA, or a National Instruments PCI-6110 board, using acquisition software custom-written in C++/MATLAB incorporating modules from Ephus (http://www.ephus.org)[104] for recordings performed in the UK. Custom stimulating electrodes were

manufactured using twisted Ni:Cr wire. For TC fibre stimulation, stimulation electrodes were inserted into VB thalamus. Stimulation in VB thalamus provides monosynaptic input to layer 4, and with single stimuli evokes direct excitatory and FFI input to layer 4 SCs. Biphasic, 100 μs constant voltage pulses were delivered by an optically isolated stimulus generator (AMPI systems) at 0.03 Hz inter-pulse frequency. Layer 4 SCs were recorded in barrels that showed >100 μV TC-evoked field potentials. Cells were selected for recording using DIC optics under infrared illumination based upon somatic morphology and laminar position. Resting potential was measured in bridge balance configuration immediately after breaking in. Recording quality was monitored on-line incorporating the following criteria: resting potentials more hyperpolarized than −50 mV, stable (<25 MΩ, <20% drift) access resistance and holding currents <100 pA. Cell-attached recordings were performed according to reference[105]. Briefly, stable greater than gigaohm seals were obtained between electrode and neuron with minimal suction to minimise mechanical membrane stress. Recordings were performed in voltage-clamp configuration by holding cells at pipette potentials that were empirically determined in current clamp to minimise holding current. Recordings were accepted if all action currents (spikes) fired were <−200 pA in peak amplitude, seal resistance and holding current were stable for the duration of the experiment. In our hands, the statistics of network-evoked firing responses were stable for over an hour (Figs. 1, 5, 6 and Supplementary Fig. 4 and data not shown), in contrast to a previous study[106,107]. For recordings from putative FS interneurons, we targeted neurons with large, cigar-shaped somata in the barrel hollow close to the bottom of L4. Since multiple electrophysiologically defined classes of FS interneurons have been reported in barrel cortex[108], careful inspection of firing patterns was performed before further analysis. Only neurons firing early in current step, with a non-stuttering firing profile were included for analysis in this study (Fig. S1a). Simultaneous whole-cell recordings from two neurons ('paired recordings') co-located within the same barrel were performed in the same manner. Synaptic connectivity was tested by injecting a suprathreshold current step into the presynaptic neuron whilst monitoring synaptic currents in the postsynaptic neuron in current clamp at holding potentials of either −70 mV (for Glutamatergic synapses) or 0 mV (GABAergic synapses). A minimum of 20 trials were used to ascertain connectivity.

**Impedance profiling.** Sinusoidal current waveforms (ZAPs), with $I_{inject}(t) = A\sin(2\pi f t)$ were injected into current-clamped neurons, with amplitude (A) of ±40 pA and frequency (f) increasing between 0.5 and 50 Hz over 25 s. Fourier-transformed voltage and current waveforms were used to derive the neurons' complex impedance $\bar{Z}(f)$ according to $\bar{Z}(f) = \frac{\bar{V}(f)}{\bar{I}_{inject}(f)}$[109–112], where superscripts denote the Fourier operator[50]. The impedance magnitude, $|\bar{Z}(f)|$ is thus a real valued number between $R_{membrane}$ for sustained DC current (i.e. Ohmic resistance, $\bar{Z}(f = 0) = R_{membrane}$) and zero, decreasing to zero in the limit of $f \to \infty$. The phase shift $\phi_{(f)}$ between the input current and voltage output was taken as the inverse tangent of the ratio between the real and imaginary components of the complex impedance:

$$\phi_{(f)} = \angle \bar{Z}_{(f)} = \arctan\left(\frac{\bar{Z}_{(f)imaginary}}{\bar{Z}_{(f)real}}\right) \quad (1)$$

For Bode filter analysis of impedance profiles, the frequency-dependent system gain is expressed as log-power ratios of impedance at higher frequencies relative to that at the lowest tested (f = 0.5 Hz). Little additional output attenuation compared with DC is expected for either genotype at 0.5 Hz, i.e. where $R_{membrane} \approx Z_{(f=0.5Hz)}$, hence this is a fair normalization strategy[113]. Model first-order RC low-pass filter responses were assumed as:

$$Z = \frac{1}{\sqrt{1/R_{membrane} + i\omega C_{membrane}}}, \text{where } \omega = 2\pi f \quad (2)$$

The bandwidth, or cut-off frequency $f_{cut-off}$, of an ideal first-order low-pass filter, above which the output power relative to DC is attenuated by greater than −3 dB (i.e. approximately halved) is given as:

$$f_{cut-off} = \frac{1}{2\pi\tau_{membrane}} = \frac{1}{2\pi RC} \quad (3)$$

where here R and C are $R_{membrane}$ and $C_{membrane}$, respectively. Above $f_{cut-off}$ the voltage response decays with a frequency-dependent roll-off of −20 dB/decade. It can therefore be expected that an increase in either $R_{membrane}$ and $C_{membrane}$ will, by increasing the membrane time constant, lower the cut-off frequency of the filter effect.

**Oscillatory spike locking.** To find firing threshold, current-clamped neurons were first depolarized by bias current injection until repetitive action potential firing was observed. Holding potentials were then corrected to 5 mV hyperpolarized from this value at the start of each trial ($V_{Baseline}$[106]). Sinusoidal current stimulation was repeated five times for each frequency. To avoid affecting AP initiation by fluctuating voltage level, cells in which $V_{Baseline}$ changed by >2 mV were discarded. Phase-locking of AP firing was assessed by registering each AP's peak time to the phase of the injected current waveform for each cycle, calculating using the Hilbert transform in MATLAB.

**Data analysis**. All data were analysed using custom-written routines in MATLAB, except for additional toolboxes detailed below. Where beneficial to improve analysis speed, C++ code was compiled and called from within MATLAB as MEX files.

**Statistical analysis**. Data is shown summarised as mean ± SEM unless otherwise stated. Individual parameter distributions were tested for normality (Kolmogorov–Smirnov tests, $p > 0.05$) before comparisons were made. Non-parametric tests were used under conditions of deviation from normality, as stated in the individual figure legends.

**Spike train statistics**. For spike probability analysis, spike times from cell-attached recordings were down-sampled to 1 ms resolution and converted to spike probability functions. In this approach each spike time is discretized at 1 ms and convolved with a Gaussian kernel with a standard deviation of 1 ms. Each spike's resulting total integral was adjusted to 1 across a range of 3 standard deviations. Correspondingly, spike times are smoothed such that a spike occurs at time $t$ contributes an extra ~0.1 spike probability to that of the next spike if it arises 30 ms later. The resulting spike density convolution is a continuous function rather than a discrete train; more amenable to comparison between trials and recordings, and less sensitive to binning artefacts than with a raw spike count histogram approach.

**Thalamocortical integration simulations**. NEURON code for simulations that repeat the main findings of these results and a graphical user interface for exploring interacting parameters in a feed-forward inhibitory circuit can be found at https://github.com/apfdomanski/Thalamocortical-Synaptic-Integration-in-Fmr1-KO-cortex. A single-compartment model neuron consisting of a soma with one dendrite was instantiated in NEURON[114]. Dendrite length and global leak conductance were tuned to match genotype mean values for input resistance and whole-cell capacitance of Fmr1-KO and wild-type recordings. To recreate TC and FFI inputs, two Exp2Syn model synapses were placed at the soma, with reversal potentials of 0 mV and −71 mV, respectively. Synaptic kinetics (rise and fall times) matched voltage-clamp data. The model TC input was activated with a peak conductance of 1 nS, yielding an ~60 pA peak inward current from a leak reversal potential of −60 mV. The model FFI input was then activated with an onset latency matching voltage-clamp recordings. To recreate the effects of FFI under conditions of different G/A ratios, peak conductance that was varied on successive trials relative to the peak TC conductance between 0 and 10×, in increments of 0.5.

**Short-term plasticity model**. Short-term depression dynamics of TC-EPSC and FF-IPSC current components were quantified using bi-exponential decay fits to trial-averaged voltage-clamp recordings, normalized to initial steady-state current amplitudes. Curves in Fig. 5c are best fits to within-cell measurements of G/A ratios, normalized to steady-state G/A ratio for each cell. Least-squares optimal fits to G/A ratio depression were obtained by using the fminsearch algorithm in MATLAB to obtain terms for underlying TC-EPSC and FF-IPSC components. For implementation in NEURON (v7.3), Depressing current inputs were re-fit with a phenomenological model of STP[115]. Briefly, starting from steady-state amplitude $A_0$, the stimulus-evoked response amplitude $A$ of each current was multiplied by two dynamic depression factors, $D_1$ and $D_2$ (constrained <1):

$$A = A_0 D_1 D_2. \quad (4)$$

After each stimulus, $D_1$ and $D_2$ were multiplied by constants $d_1$ and $d_2$, representing the amount of depression per presynaptic action potential (i.e. $D_1 \rightarrow D_1 d_1$, and $D_2 \rightarrow D_2 d_2$). Between stimuli, $D$ variables recovered exponentially back towards 1 with first-order kinetics governed by recovery time constants $\tau_{D1}$ and $\tau_{D2}$:

$$\tau_{D1}\frac{dD_1}{dt} = 1 - D_1, \text{ and } \tau_{D2}\frac{dD_2}{dt} = 1 - D_2. \quad (5)$$

Accumulation of synaptic depression was therefore observed when the inter-stimulus interval was shorter than time required for recovery. Synaptic rise and decay time constants, as well as synaptic plasticity constants $d_1$, $d_2$, $\tau_{D1}$ and $\tau_{D2}$ were obtained from the genotype mean synaptic kinetics and short-term depression of EPSCs and FF-IPSCs individually. Fitting of short-term depression terms was performed on normalized synaptic depression curves for 5, 10, 20 and 50 Hz TC stimulation, equally weighted and optimised using a Levenberg-Marquardt search implemented in MATLAB.

**Layer 4 network simulations**. MATLAB code to reproduce the main findings of this paper is hosted at https://github.com/apfdomanski/Fmr1-KO-cortical-layer-4-spiking-network. A recurrently connected spiking network of single-compartment conductance-based leaky integrate-and-fire (LIF) neurons was instantiated in MATLAB. The subthreshold potential of the $i$th neuron evolved according to

$$\tau_m\frac{dv_i}{dt} = -[v_i(t) - v_{rest}] + R_m I_{tot}^i(t) \quad (6)$$

where $I_{tot}^i(t)$ is the summed synaptic current from recurrent and external inputs at time $t$. Spikes were emitted at $t^f = \{t | v(t) = v_{threshold}\}$ when $v > v_{threshold}$ leading to

the reset condition $v(t + dt) \rightarrow v_{reset} < v_{threshold}$ for a finite refractory period of 1.5 ms.

$I_{tot}^i(t)$ was calculated at each time step as:

$$I_{tot}^i(t) = \sum_{N_{(i,AMPArec)}} I_{AMPArec}^i(t) + \sum_{N_{(i,NMDArec)}} I_{NMDArec}^i(t)$$
$$+ \sum_{N_{(i,GABArec)}} I_{GABArec}^i(t) + I_{AMPArec}^i(t) + I_{NMDArec}^i(t), \quad (7)$$

with $N_{(i, AMPArec)}$, $N_{(i, NMDArec)}$ and $N_{(i, GSBArec)}$ being the set of $N$ recurrent synapses projecting onto the $i$th neuron, and $I_{AMPArec}^i(t)$, $I_{NMDArec}^i(t)$, $I_{GABArec}^i(t)$, $I_{AMPAext}^i(t)$, $I_{NMDAext}^i(t)$ the synaptic inputs from recurrent AMPA, NMDA and GABA and external AMPA and NMDA synapses, respectively.

Synaptic input current depended also on the postsynaptic membrane potential:

$$I_{syn}^i(t) = A^i(t) \cdot \overline{g_{syn}^i} \cdot s_{syn}(t)\left(v(t) - E_{syn}\right), \quad (8)$$

where $A^i(t)$ is the short-term depression state of synapse $i$ described by Eq. (4), $\bar{g}_{syn}$ and $E_{syn}$ are, respectively, the maximal synaptic conductance and reversal potential of the synapse. $s_{syn}(t)$ is a delayed-sum-of-exponentials (alpha) synapse model (after ref. [116]):

$$S_{syn}(t) = f\left(e^{-\frac{t_{conduct} - \Delta t}{\tau_{decay}}} - e^{-\frac{t_{conduct} - \Delta t}{\tau_{rise}}}\right)\mathcal{H}(\Delta t), \quad (9)$$

in which $t_{conduct}$ is a finite synaptic conduction delay, $\Delta t > -\infty$ is the elapsed time $t - t^f$ since the time of the last presynaptic spike. The Heaviside step function $\mathcal{H}$ ensures causality in time. The function $f(x)$ is an amplitude normalisation factor:

$$f(x) = \frac{1}{-e^{-(t_{peak} - t_{conduct} - \Delta t)/\tau_{rise}} + e^{-(t_{peak} - t_{conduct} - \Delta t)/\tau_{rise}}}, \quad (10)$$

where the amplitude of the conductance is maximal at time $t_{peak}$:

$$t_{peak} = \Delta t + t_{conduct} + \frac{\tau_{rise}\tau_{decay}}{\tau_{decay} - \tau_{rise}}\ln\left(\frac{\tau_{decay}}{\tau_{rise}}\right), \quad (11)$$

NMDA conductances displayed additional voltage dependence:

$$I_{NMDA}^i(t) = A^i(t) \cdot G(v(t)) \cdot \overline{g^i}_{NMDA} \cdot s_{syn}(t)\left(v(t) - E_{NMDA/AMPA}\right), \quad (12)$$

In which $G(v(t))$ describes the voltage-dependent Mg$^{2+}$-sensitive blockade of NMDA receptors:

$$G(v(t)) = \frac{1}{1 + \frac{e^{-av(t)}[Mg^{2+}]_{out}}{b}}, \quad (13)$$

with $a = 0.062$ mV$^{-1}$ and $b = 3.57$ mM[117,118] and using our experimental extracellular Magnesium concentration $[Mg^{2+}]_{out}$ of 1.3 mM.

**Network architecture**. 800 Excitatory and 150 Inhibitory neurons were randomly connected with probabilities obtained experimentally in Fig. 2, excluding autapses. All classes of synaptic connection were randomly assigned fractional weights up to $\bar{g}_{syn}$, with the exception of Ex–Ex connections, which were log-normally distributed in line with our experimental findings (Fig. 3e) and previous reports[33,43,119]. Simulated external thalamocortical inputs were provided to 80% of Ex and In neurons. $\bar{g}_{syn}$ of external NMDA and AMPA were jittered between trials and thalamorecipient neurons were shuffled to introduce variability between trials. Short-term plasticity at TC synapses was modelled using the same fits to data as described above. Simulations (1000 ms) were carried out using forward Euler integration at 0.5 ms resolution. Spike times were discretized to 1 ms resolution. Modelling results show responses of 800 Ex neurons across $N = 10$ network simulation trials for each stimulation condition each from five random network seeds. For ensemble decoding analysis shown inf Fig. 10, results are calculated for 10 random permutations of each ensemble size, four oddball stimulus positions. Simulations were performed on five random network seeds, ten trials for each oddball position and frequency combination.

**Parameter distributions**. Cell-intrinsic, synaptic and connectivity parameters used are detailed in Supplemental Table 1. Where two numbers are shown, they represent mean and standard deviation of jittered variables. All parameters were assumed to vary independently, excluding input resistance and membrane capacitance, which displayed both private and shared variance.

**Spike train metrics and ensemble decoding**. Spike train distance metrics were computed with the method described by van Rossum[66] with a decay time constant of 50 ms. We used an optimised calculation of the van Rossum metric implemented in C++ by Houghton and Kreuz[120]. Ensemble decoding was performed using a multivariate linear decoder using ensemble spike trains as covariates. To prevent over-fitting, pooled covariance matrices for regular and oddball conditions were regularised by a factor of 0.05 on the identity matrix. Decoder accuracy was evaluated with tenfold leave-one-out cross-validation, with decoders trained on nine trials asked to predict the stimulus condition that lead to the outcome of the

remaining withheld trial. Training/test sets were split 70/30% and independent evaluation sets were used.

**Reporting summary**. Further information on research design is available in the Nature Research Reporting Summary linked to this article.

## Data availability
All raw data generated/analysed in this study are deposited as Supplementary Material on the Nature Publishing Group website.

## Code availability
NEURON code for the single-cell model and MATLAB/C++ code for the Layer 4 network model are publicly available on Github via the above repository links. MATLAB/C++ code for analysis of data and simulations will be available from the authors upon reasonable request.

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

## Acknowledgements

The authors wish to thank Ramesh Chittajallu, Chris McBain, Michael Ashby, Michael Daw, Timothy o'Leary, Mark van Rossum and past/current members of the Kind, McBain, Isaac and Wyllie laboratories for invaluable scientific input, and Owen Dando and Zrinko Kozic for statistical discussions. Funders: Simons Foundation Autism Research Initiative (529085), The Patrick Wild Centre, Medical Research Council UK (MR/P006213/1), The Shirley Foundation and the RS Macdonald Charitable Trust, Wellcome-National Institutes of Health Collaborative Scholarship (APFD).

## Author contributions

A.P.F.D.: designed and interpreted, performed experiments, developed and deployed simulations, analysed data and wrote the paper; S.A.B.: designed and performed experiments, analysed/interpreted data; D.J.A.W.: designed experiments, analysed/interpreted data; J.T.R.I.: designed experiments, analysed/interpreted data, obtained funding and wrote the paper P.C.K.: designed experiments, analysed/interpreted data, obtained funding and wrote the paper.

## Competing interests

The authors declare no competing interests.
