## [Peer Review File · Nature Communications]

Reviewers' Comments:

Reviewer #1:

Remarks to the Author:

This is an exciting study that combines electrophysiology and modeling approaches to examine circuit excitability defects in the barrel cortex of Fmr1 KO mice, the Fragile X mouse model. By systematically studying cellular and synaptic properties of layer 4 neurons and combining with cellular and network simulations, they found that elevated intrinsic excitability observed in Fmr1-KO SCs is the dominating feature driving the changes in spike output and impaired coding. However, the changes in the Inhibitory/Excitatory (In/Ex) balance and short-term plasticity (STP) did not result in an increase in cellular excitability but instead acted to stabilize synaptically-driven output of the neuron. Authors proposed that neurons alter their In/Ex balance to compensate for other changes in the neuron's physiological properties, and thus this compensation has a distinct contribution to the overall hyperexcitability phenotype.

The study is very comprehensive and spans from analyses of synaptic and cellular defects to circuit processing. Experiments are very well designed, results are solid and convincing. The paper is well written and of a very high quality. This work makes a critical step forward in deciphering the mechanisms and implications of circuit hyperexcitability in Fragile X. I strongly support publication, but have a few suggestions to improve the ms.

1. My major suggestion is related to the unnecessary conclusion of what constitute primary vs. compensatory changes. The discussion states that "Many of the cellular phenotypes described here have opposite effects on excitability on circuit output, strongly suggesting that some, if not all, are likely compensatory changes driven downstream of the direct effects of FMRP deletion". This study provides no evidence to support this conclusion, and in my view it only weakens the paper. There are multiple reasons why this can be incorrect, or at the very least cannot be deduced from the current set of experiments.

Here just a few considerations: The four groups of neuronal/synaptic properties that are considered in the modeling are not independent, but intimately interrelated. Figuring out which one is a primary cause and which one is a consequence is a tall order, way beyond the current data. For example, intrinsic excitability, synaptic strength and STP are all interdependent via common dependence on intracellular calcium levels. For example, an increased calcium-dependent form of STP (augmentation) in hippocampal Fmr1 KO neurons arises from altered AP properties, which is in turn caused by direct interaction of FMRP with an ion channel. So are these all primary defects, or compensatory? There is an additional issue that some of these changes only manifest themselves under certain conditions. The synaptic strength and excitability/AP firing may manifest as normal at baseline, but abnormally elevated during high-frequency bursts. Conclusions on what is perceived as primary/compensatory may thus depend on experimental protocol, but does not however make the observed effects less or more directly related to FMRP loss.

Also the way this point is presented, it makes an (unintended?) impression that the output spiking of Layer 4 feed-forward circuit in the barrel cortex is an established principle disease causing phenotype. From this perspective, synaptic and cellular changes are seen to be compensatory if they are antagonistic in modulating Layer 4 circuit function. The conclusion made from this viewpoint is that elevated intrinsic excitability of SCs is a dominate primary feature driving circuit defects, while other synaptic and cellular changes are compensation in response to this pathology acting to limit circuit dysfunction. This point is based entirely on assumption that the FFI circuit output is the principle cause of the disease state, which is not known. Moreover, it is possible to envision that there could be multiple contributing factors that act independently or in parallel with excitability changes and are

relevant to FXS phenotypes. This does not preclude these changes from being antagonistic from the perspective of excitability. Many may argue that it is LTP/LTD changes which are ultimately important and altered spiking output is a compensation, or maybe it is actually glia dysfunction which is most relevant and the spiking changes is a compensation. The current study does not have the necessary information and does not need to address this issue.

2. In addition to the question of perspective, whatever is seen as a compensation is based solely on a modeling approach using a good but a simplified 2D model and a small selected number of parameters. Moreover, authors do not go into detailed enough analysis to really figure out which of many changes are causal or compensatory; for example STP changes seem to go well with the spiking output changes, does it make STP changes primary and not compensatory? Some limitations of modeling need to be acknowledged more clearly, i.e. that modeling was performed in a single-compartment model, without a detailed 3D neuronal structure reconstruction and thus offers a first-order approximation. Strong conclusions which are relying exclusively on this modeling are overly optimistic. More work can and need to be done in the future, for example some of the “rescue” results can be verified using dynamic clamp. Although this is probably beyond the scope of the current ms, a discussion of this point is needed.

3. Unless I missed it, control for determining relative contribution of feed-forward inhibition vs direct inhibition in Fig 3 is missing.

4. Many findings on changes in FFI circuit presented here are in agreement with another study of FFI circuit in the hippocampus (Wahlstrom-Helgren and Klyachko, J Physiol 2015), which is not cited or discussed.

5. The conclusions on “paradoxical” changes in E/I ratio and resulting changes in FFI circuit output might need to be more nuanced or rephrased because the interrelationship between E/I ratio and changes in the FFI circuit output is rather complex (Wahlstrom-Helgren and Klyachko, J Neurophysiol 2016) and current observations are not necessarily paradoxical.

6. Discussion, section on “Abnormal intrinsic neuronal membrane properties”. In many brain areas, including layer 2/3 prefrontal cortex (Routh et al, J Physiol 2017), Layer 2/3 entorhinal cortex (Deng and Klyachko, Cell Reports 2016), and CA3 area of hippocampus (Deng and Klyachko J Neurosci 2018), neurons do not have changes in input resistance in Fmr1 KOs, but nevertheless have altered excitability and spiking.

7. There are some discrepancies in the data that need clarification:

Figure 1A,B shows the altered intrinsic properties of SCs, which contribute to increased number of AP fired in response to depolarization in Fmr1 KO mice. In contrast, the same altered intrinsic properties of FS interneurons (Figure 1E,F) caused a decrease in number of AP fired.

Also, how do both increased number of AP fired in SCs (Figure 1B) and decreased number of AP fired in FS interneurons (Figure 1F) caused reduced instantaneous AP frequency (Figure 1C and 1G).

Excitability: the results in Figure 4A (KO data) seem to contradict Figure 5A.

Firing probability: data in Figure 5B,C seem to contradict Figure 6A,B,C.

Firing probability: data in Figure 6A,B,C seem to contradict Figure 7B,C and 9C.

Reviewer #2:

Remarks to the Author:

Tactile reactivity (hyposensitivity, hypersensitivity) is common characteristic of neurodevelopmental disorders, yet the underlying cellular mechanisms are not well known. Genetically altered mouse models provide unique means to study cellular and molecular mechanisms of tactile reactivity. A particular advantage of the mouse model in such studies is that the nocturnal rodents such as the mouse have a highly conspicuous, patterned representation of their tactile organ, the whiskers on the snout, in their primary somatosensory cortex, known as the barrel cortex. In this study the authors take advantage of this excellent model system to investigate intrinsic membrane properties of layer 4 barrel cells and their synaptic connectivity with the thalamus in *Fmr1* knockout mice, an established genetic model for the Fragile X Syndrome (FXS). The study is well-designed and executed with appropriate controls. They carefully detail the cellular and synaptic aberrations and possible underlying causes of impaired sensory coding at the thalamocortical synapse. Importantly, they note altered excitation/inhibition (E/I) balance that could account for somatosensory or tactile hypersensitivity. A pioneering study on the excitability profiles of cortical neurons in a neurodevelopmental mouse model for the Rett syndrome (*Mecp2*-mutant mice) was done by Sacha Nelson and colleagues (Dani et al., *PNAS*, 2005, Aug 30;102(35):12560-5). By using whole-cell patch-clamp recordings from layer 5 (L5) pyramidal neurons in barrel cortex slices, they found reduced spontaneous firing which was not accompanied by a change in the intrinsic excitability of the L5 neurons but that the excitatory synaptic drive onto them was decreased and the inhibitory drive increased. They showed that the E/I ratio shifted to favor inhibition over excitation in *Mecp2*-mutant mice. They concluded that such shifts in E/I balance could underlie some of the RTT-specific symptoms. This study is overlooked and not cited. More importantly, citation and discussion of highly relevant recent work from Lo and Erzurumlu is also missing. These authors used similar whole-cell recordings from thalamocortical slices and investigated the properties of layer 4 barrel cortex cells and their thalamocortical synaptic connectivity in *Mecp2*-mutant and another autism spectrum disorder (ASD) model mouse *Met-Emx1* (Lo et al., *J Neurophysiol.* 2016 Mar;115(3):1298-306; Lo et al., *J Neurosci.* 2016 Mar 30;36(13):3691-7). They found shifts in E/I balance in different directions in each of these mutants in the barrel cortex. In *Met-Emx1* mice, decreased GABAA receptor-mediated inhibition shifts the E/I balance toward excitation in the barrel cortex. In a recent work, they investigated insulin sensitivity and found that application of an insulin sensitizing agent to the thalamocortical slices restores the E/I balance (Lo and Erzurumlu, *Mol Autism.* 2018 Feb 22; 9:13.) These findings are highly relevant to the Domanski et al., study here as each of these studies document specific cellular and synaptic defects in the barrel cortex. It is important to compare and contrast synaptic circuitry defects in the primary somatosensory cortex of model mice as in various neurodevelopmental disorders in humans the tactile sensitivity is altered in either direction during development and could underlie various cognitive and social behaviors that develop later on.

One of the cited studies here (Deng et al., *Neuron.* 2013 Feb 20;77(4):696-711) showed that FMRP regulates neurotransmitter release in CA3 hippocampal neurons and its loss leads to excessive neurotransmitter release. The study seems to lack experiments on presynaptic release properties of thalamocortical axons synapsing with layer 4 cells, and their potential role on hypersensitivity.

Other comments:

Fifteen years ago, Rubenstein and Merzenich published an influential review (*Genes Brain Behav.* 2003 Oct;2(5):255-67), in which they suggested that neurodevelopmental disorders like ASD reflect an increase in the ration of excitation and inhibition. Since then the concept of Excitation/Inhibition has become established, and abbreviated as E/I balance (also see Nelson and Valakh, *Neuron*, 2015 Aug 19; 87(4): 684–698.) For some unknown reason, the authors have come up with an unconventional

nomenclature and use "Inhibitory/Excitator" or "In/Ex"; "ex" as a preposition or noun has other connotations, they should change their terminology to the conventional one and use "E/I" balance throughout the paper.

Somatosensation plays an important role in development. Neurodevelopmental disorders, including Fragile X, comprise a broad range of conditions that also include abnormal "tactile reactivity." The authors, based on their study, in general refer to and use the term "hypersensitivity." They should be more inclusive and consider the overall picture and keep in line with the developing new terminology in the field and use "tactile reactivity," or "tactile responsiveness" instead. Some of the neurodevelopmental disorders are characterized by tactile hyposensitivity. In this context, I refer the authors to a review by Cascio J Neurodev Disord. 2010 Jun; 2(2): 62-69.

The pages are not numbered but in second paragraph of the page that has Introduction, lines 4-6 should be revised if "known cellular processes" are numerous, "even greater potential number" doesn't make sense.

Reviewer #3:

Remarks to the Author:

General comments:

This manuscript by Domanski and others explore how developmental changes alter circuit balance underlying sensory hyperexcitability in FXS. Cellular and synaptic recordings were performed in layer 4 of the barrel cortex in Fmr1 knockout mice, and circuit pathology was studied. A combination of in vitro and in silico experiments reveal how variations in phenotypes at key developmental time points result in sensory processing dysfunction. Overall this is an excellent paper that is important for our understanding of circuit plasticity as well as how we understand how subtle circuit changes that are observed in neurodevelopmental disorders can result in substantial pathophysiological problems. Overall, this is a worthwhile study and contains some very important information. However, the manuscript requires revision and a re-assessment of data analysis and interpretations.

1. Concerns

Major

1) The recordings are of high quality but how do the authors know they are recording from "paired" whole-cell patches? Please provide more explanation.

2) How do the authors know they are recording from specific cell types? They state, "Cells were selected for recording using DIC optics under infrared illumination based upon somatic morphology and laminar position DIC optics under infrared illumination based upon somatic morphology and laminar position". Current clamp traces are shown, however there are subtle variations of layer 4 fast spiking cells in the barrel cortex (Li and Huntsman 2014 Neuroscience). These variations include whether there is a delay in the first action potential and whether there is accommodation of action potential firing. The differences have biophysical significance in that rheobase and input resistance are different. The authors should provide intermediate traces to show these fast spiking cells are the same.

3) Is only looking at P10-11 timepoint sufficient? Would it be useful to look at older time points and younger with the same paradigms? How do the changes at this age P10/11 explain alterations seen at P14 or adult animals in Fmr1 Knock out animals? Please provide an explanation.

4) The terminology is a little confusing. Do people commonly express these things as G/A ratio? Therefore also In/Ex is it commonly expressed this way? Most studies use E/I balance.

Minor

1)The authors could really benefit by providing a summary figure.

2) The reference manager that was used resulted in variations of the literation cited in the text.

Reviewer #4:

None

Response to reviewer's comments

We would like to thank the reviewers for their positive comments on the "quality" and "comprehensiveness" of our manuscript. We would also thank them for their helpful comments which we have addressed in the revised manuscript which we believe has improved it significantly. Below, we have addressed each of their comments (italics). Where a particular comment has been subdivided to address , we have added subheadings in bold (e.g. **a, b, c etc.**).

Reviewer #1 (Remarks to the Author):

1(a) My major suggestion is related to the unnecessary conclusion of what constitute primary vs. compensatory changes. The discussion states that "Many of the cellular phenotypes described here have opposite effects on excitability on circuit output, strongly suggesting that some, if not all, are likely compensatory changes driven downstream of the direct effects of FMRP deletion". This study provides no evidence to support this conclusion, and in my view it only weakens the paper. There are multiple reasons why this can be incorrect, or at the very least cannot be deducted from the current set of experiments.

We thank the Reviewer for the very helpful and constructive criticism. Having reinterpreted our experimental and modeling results we agree that our previous conclusions on the direct vs. compensatory nature of the changes reported in the current study were too strong and not fully supported by the current data. We have majorly overhauled our discussion section on this point. Moreover, we now include suggestions about how to further approach the dissection of these effects with experiments and theory. We have also altered the title of the manuscript. Our specific points in rebuttal are interdigitated below.

1(b) Here just a few considerations: The four groups of neuronal/synaptic properties that are considered in the modeling are not independent, but intimately interrelated. Figuring out which one is a primary cause and which one is a consequence is a tall order, way beyond the current data. For example, intrinsic excitability, synaptic strength and STP are all interdependent via common dependence on intracellular calcium levels.

One of the motivations for using grouped parameters in the simulations was to be agnostic to the potential common underlying mechanisms. We now reference studies in which synaptic E/I tone and intrinsic excitability have been shown to change independently in other reports on pathophysiology in mouse lines of ASD-related genes and are also explicit on this point in discussing the limitations of the modeling approach used.

1(c) For example, an increased calcium-dependent form of STP (augmentation) in hippocampal Fmr1 KO neurons arises from altered AP properties, which is in turn caused by direct interaction of FMRP with an ion channel. So are these all

primary defects, or compensatory? There is an additional issue that some of these changes only manifest themselves under certain conditions. The synaptic strength and excitability/AP firing may manifest as normal at baseline, but abnormally elevated during high-frequency bursts. Conclusions on what is perceived as primary/compensatory may thus depend on experimental protocol, but does not however make the observed effects less or more directly related to FMRP loss.

We agree that our original manuscript lacked precision in the discussion of primary vs. compensatory changes with respect to information transmission. We previously presented this point in relation to the rapid change to steady-state E/I balance during repetitive stimulation during physiological input patterns. Although we cannot isolate the primary mechanism here, we have now strengthened this discussion point by referencing classic papers relating information transmission and temporal filtering to short-term plasticity (Tsodyks and Markram 1998) as well as more recent studies on information transmission alternations of relevance from Deng and Klyachko.

1(d) Also the way this point is presented, it makes an (unintended?) impression that the output spiking of Layer 4 feed-forward circuit in the barrel cortex is an established principle disease causing phenotype. From this perspective, synaptic and cellular changes are seen to be compensatory if they are antagonistic in modulating Layer 4 circuit function. The conclusion made from this viewpoint is that elevated intrinsic excitability of SCs is a dominant primary feature driving circuit defects, while other synaptic and cellular changes are compensation in response to this pathology acting to limit circuit dysfunction. This point is based entirely on assumption that the FFI circuit output is the principle cause of the disease state, which is not known. Moreover, it is possible to envision that there could be multiple contributing factors that act independently or in parallel with excitability changes and are relevant to FXS phenotypes. This does not preclude these changes from being antagonistic from the perspective of excitability. Many may argue that it is LTP/LTD changes which are ultimately important and altered spiking output is a compensation, or maybe it is actually glia dysfunction which is most relevant and the spiking changes is a compensation. The current study does not have the necessary information and does not need to address this issue.

We thank the reviewer for pointing this out. We have limited our discussion point of the primacy of Layer 4 dysfunction explicitly stating that *in vivo* correlates are yet to be confirmed and that contributions from intrinsic excitability is one underlying contribution, potentially deriving from multiple potential mechanisms.

2. In addition to the question of perspective, whatever is seen as a compensation is based solely on a modeling approach using a good but a simplified 2D model and a small selected number of parameters. Moreover, authors do not go into detailed enough analysis to really figure out which of many changes are causal

or compensatory; for example STP changes seem to go well with the spiking output changes, does it make STP changes primary and not compensatory? Some limitations of modeling need to be acknowledged more clearly, i.e. that modeling was performed in a single-compartment model, without a detailed 3D neuronal structure reconstruction and thus offers a first-order approximation. Strong conclusions which are relying exclusively on this modeling are overly optimistic. More work can and need to be done in the future, for example some of the “rescue” results can be verified using dynamic clamp. Although this is probably beyond the scope of the current ms, a discussion of this point is needed.

We acknowledge the limitations of our modeling approach and agree with the reviewer that we need to be more transparent on this concern. To address this criticism explicitly we have included a substantial section in the discussion expanding on this point, suggesting opportunities both for further use of the model in a dynamic clamp setting as well as extensions to full 3D simulations.

3. Unless I missed it, control for determining relative contribution of feed-forward inhibition vs direct inhibition in Fig 3 is missing.

We believe the reviewer has misinterpreted our methodology here: in the present study the isolated stimulation presented in ventrobasal thalamus only activates thalamocortical input to the cortex, thus in response to single stimulus only evoked feed-forward inhibition is observed in Layer 4. We have expanded on the description of thalamic stimulation in the methods section to clarify this.

4. Many findings on changes in FFI circuit presented here are in agreement with another study of FFI circuit in the hippocampus (Wahlstrom-Helgren and Klyachko, J Physiol 2015), which is not cited or discussed.

We thank the reviewer for pointing out this omission and have included references to these studies in a discussion section drawing parallels between the present and previous literature on altered FFI in Fmr1-KOs.

5. The conclusions on “paradoxical” changes in E/I ratio and resulting changes in FFI circuit output might need to be more nuanced or rephrased because the interrelationship between E/I ratio and changes in the FFI circuit output is rather complex (Wahlstrom-Helgren and Klyachko, J Neurophysiol 2016) and current observations are not necessarily paradoxical.

We have rephrased these conclusions strengthening reference to the dynamic circuit behaviour and further promote the role of dynamic data modeling in dissecting the complex circuit interactions between E/I balance and circuit excitability.

6. Discussion, section on “Abnormal intrinsic neuronal membrane properties”. In

many brain areas, including layer 2/3 prefrontal cortex (Routh et al, J Physiol 2017), Layer 2/3 entorhinal cortex (Deng and Klyachko, Cell Reports 2016), and CA3 area of hippocampus (Deng and Klyachko J Neurosci 2018), neurons do not have changes in input resistance in Fmr1 KOs, but nevertheless have altered excitability and spiking.

We thank the reviewer for pointing this out and have expanded on this point in the discussion of this section.

7(a). *There are some discrepancies in the data that need clarification: Figure 1A,B shows the altered intrinsic properties of SCs, which contribute to increased number of AP fired in response to depolarization in Fmr1 KO mice. In contrast, the same altered intrinsic properties of FS interneurons (Figure 1E,F) caused a decrease in number of AP fired.*

Although in the course of this study we have not isolated the underlying mechanism here we do not feel that this alters our conclusions. We attribute this effect to cell-type specific alterations to voltage-dependent conductances in the SC and FS cells, manifesting as a broader AP in both cell types: Compared to WT SC neurons, AP repolarization speed was slower in the *Fmr1-KO* SCs. In contrast, *Fmr1-KO* FS neurons displayed reductions in both spike depolarization and repolarization speed (data not shown). We thus hypothesize that these extra effects in FS neurons offset the elevated input resistance and impair their ability to rapidly fire trains of APs. We have clarified this point in the results.

7(b). *Also, how do both increased number of AP fired in SCs (Figure 1B) and decreased number of AP fired in FS interneurons (Figure 1F) caused reduced instantaneous AP frequency (Figure 1C and 1G).*

This is due to methodology and the differences between cells in their spike rate accommodation: The spike counts (Fig 1B,F) were performed over a fixed window of 500ms, whereas the ISIs were calculated after letting the cells fire up to the displayed spike counts for each cell type (abscissae, Figures 1C,G). The SCs typically fire <10 spikes and then stop, whereas at twice-rheobase the FSs continue firing for the duration of the current step. The discrepancy is due to only the first 10 spikes being analysed for the SC case, and the marked spike rate accommodation visible in the initial firing. Thus the effect is more compressed into the start of the spike train and drops away after approx. 3 spikes. We have clarified this point in the results text for Figure 1.

7(c). *Excitability: the results in Figure 4A (KO data) seem to contradict Figure 5A.*

This is resolved by comparing the relative kinetics of short-term plasticity of the different input currents to SCs: although these current ratios typically start from an elevated level in the KOs, the rundown of FF-inhibitory tone is faster and more

progressive in KO SCs than that of thalamic EPSCs, which show slower and more gradual depression (group data in Figure 4C). Coupled with the increased intrinsic excitability of the KO neurons, this produces an increase in voltage summation and contributes to ectopic spiking. We have changed the description of this effect to make this more explicit in the results text.

7(d). *Firing probability: data in Figure 5B,C seem to contradict Figure 6A,B,C.*

Data in Fig 5 covers a range of physiological stimulation frequencies, whereas Fig 6 exclusively uses 50Hz stimulation. The data at 50Hz for the two figures are comparable: please note that the spike density measurement employed encapsulates both the changes to the spike rate and timing, as well as the reduced inter-trial fidelity of the response train. This is further broken down in Figure S5A,B. We have clarified this in the results text and in the legend of Figure 6.

7(e). *Firing probability: data in Figure 6A,B,C seem to contradict Figure 7B,C and 9C.*

The model presented in Figure 7 lacks the recurrently connected architecture of the real LIV circuit, thus is useful to disentangle the contributions of direct vs. feedback processing on the firing patterns of the SCs. The observation that the KO model in Figure 7 fires spikes over a larger range of stimulus conditions, but with delayed and variable onset isolates the contribution of distorted feed-forward thalamic processing to this effect and is consistent with that of Figure 6 (Firing of KO neurons delayed and more variable compared to WT counterparts). [We have highlighted in the legend for Figure 7B that parameter ranges shown in red *extend* those shown in blue, i.e. for all the WT conditions, the KO is also firing]. Moreover, the reduced spike counts in the KO model (Figure 7C, lower panel) can be attributed to a combination of slower current kinetics, longer membrane time-constant and more progressive STP of excitatory input currents included in the model parameter space. This is further dissected in Figure 8. Figure 9C reflects that once the recurrent architecture of LIV is included into the model of thalamocortical response, the circuit hyperexcitability is revealed. Together with Figure 7, this shows that the effects of TC hyper-responsiveness in the KOs lead to an exaggerated recruitment of network firing, as demonstrated in Figure 6. Figure 9C presents an integrated measure of spike count/response variability. We have clarified the relative merits of the ‘thalamic input only’ and ‘recurrent architecture models’ (i.e. Figures 7-8) in the discussion.

Reviewer #2 (Remarks to the Author):

1a) *A pioneering study on the excitability profiles of cortical neurons in a neurodevelopmental mouse model for the Rett syndrome (Mecp2-mutant mice) was done by Sacha Nelson and colleagues (Dani et al., PNAS, 2005, Aug 30;102(35):12560-5). By using whole-cell patch-clamp recordings from layer 5 (L5) pyramidal neurons in barrel cortex slices, they found reduced spontaneous*

firing which was not accompanied by a change in the intrinsic excitability of the L5 neurons but that the excitatory synaptic drive onto them was decreased and the inhibitory drive increased. They showed that the E/I ratio shifted to favor inhibition over excitation in Mecp2-mutant mice. They concluded that such shifts in E/I balance could underlie some of the RTT-specific symptoms. This study is overlooked and not cited.

We thank the Reviewer for pointing out this oversight and have added the Dani et al study to the discussion section addressing independent effects on intrinsic and synaptic dysfunction.

1b *More importantly, citation and discussion of highly relevant recent work from Lo and Erzurumlu is also missing. These authors used similar whole-cell recordings from thalamocortical slices and investigated the properties of layer 4 barrel cortex cells and their thalamocortical synaptic connectivity in Mecp2-mutant and another autism spectrum disorder (ASD) model mouse Met-Emx1 (Lo et al., J Neurophysiol. 2016 Mar;115(3):1298-306; Lo et al., J Neurosci. 2016 Mar 30;36(13):3691-7). They found shifts in E/I balance in different directions in each of these mutants in the barrel cortex. In Met-Emx1 mice, decreased GABAA receptor-mediated inhibition shifts the E/I balance toward excitation in the barrel cortex. In a recent work, they investigated insulin sensitivity and found that application of an insulin sensitizing agent to the thalamocortical slices restores the E/I balance (Lo and Erzurumlu, Mol Autism. 2018 Feb 22; 9:13.) These findings are highly relevant to the Domanski et al., study here as each of these studies document specific cellular and synaptic defects in the barrel cortex.*

We have added the Lo et al papers to the cited literature with regards to efforts to normalized E/I balance.

2) *It is important to compare and contrast synaptic circuitry defects in the primary somatosensory cortex of model mice as in various neurodevelopmental disorders in humans the tactile sensitivity is altered in either direction during development and could underlie various cognitive and social behaviors that develop later on.*

We agree with this point and have included extensive comparisons to the literature on older animals in the discussion section. Please also see our responses to Reviewer 3 on the same point.

3) *One of the cited studies here (Deng et al., Neuron. 2013 Feb 20;77(4):696-711) showed that FMRP regulates neurotransmitter release in CA3 hippocampal neurons and its loss leads to excessive neurotransmitter release. The study seems to lack experiments on presynaptic release properties of thalamocortical axons synapsing with layer 4 cells, and their potential role on hypersensitivity.*

Our results on short-term plasticity at thalamic input to SCs, and at synapses between pairs of connected SC and FS neurons show synapse-specific

increases in the rate of depression in *Fmr1-KO* recordings that affect the temporal evolution in the Ex/In balance during stimulation. Although we do not dissect these results further the most likely explanation for these effects is an alteration in presynaptic function. Moreover, separation of possible presynaptic mechanisms (e.g. FMRP regulation of quantal release probability and/or failure rate) would not alter our circuit-level interpretations of the role that short-term plasticity plays in the overall network level dysfunction. We have added these points to the discussion section on short-term plasticity.

4) *Fifteen years ago, Rubenstein and Merzenich published an influential review (Genes Brain Behav. 2003 Oct;2(5):255-67), in which they suggested that neurodevelopmental disorders like ASD reflect an increase in the ration of excitation and inhibition. Since then the concept of Excitation/Inhibition has become established, and abbreviated as E/I balance (also see Nelson and Valakh, Neuron, 2015 Aug 19; 87(4): 684–698.) For some unknown reason, the authors have come up with an unconventional nomenclature and use “Inhibitory/Excitator” or “In/Ex”; “ex” as a preposition or noun has other connotations, they should change their terminology to the conventional one and use “E/I” balance throughout the paper.*

We thank the reviewer for raising this point that helps our study align better to the established literature. To clarify, our use of the term “In/Ex” balance throughout in the submitted manuscript was motivated by the technical definition of GABA/AMPA (abbreviated to G/A) ratio used to describe the evoked GABAergic inhibition received by L4 neurons during thalamic stimulation, as normalized by the strength of driving excitation (predominantly AMPA mediated). We concede that in generalizing this term to In/Ex throughout the manuscript we have introduced new and potentially confusing nomenclature and thus have changed all instances of “In/Ex” to “E/I” in the revised manuscript. We have also clarified this point in the text and have added reference to the Nelson and Valakh review.

3) *Somatosensation plays an important role in development. Neurodevelopmental disorders, including Fragile X, comprise a broad range of conditions that also include abnormal “tactile reactivity.” The authors, based on their study, in general refer to and use the term “hypersensitivity.” They should be more inclusive and consider the overall picture and keep in line with the developing new terminology in the field and use “tactile reactivity,” or “tactile responsiveness” instead. Some of the neurodevelopmental disorders are characterized by tactile hyposensitivity. In this context, I refer the authors to a review by Cascio J Neurodev Disord. 2010 Jun; 2(2): 62–69.*

Although we have cited the Cascio review we had inadvertently used the older nomenclature in our introduction/discussion text and have altered our terminology accordingly in the revised manuscript. However, where we refer specifically to FXS, we maintain the term hypersensitivity.

4) The pages are not numbered but in second paragraph of the page that has Introduction, lines 4-6 should be revised if “known cellular processes” are numerous, “even greater potential number” doesn’t make sense.

We have edited this point for clarity.

Reviewer #3 (Remarks to the Author):

1. Concerns

Major

1) The recordings are of high quality but how do the authors know they are recording from “paired” whole-cell patches? Please provide more explanation.

We refer to our dual, simultaneous whole-cell recordings as ‘paired recordings’ in the text. We have added to the methods description to clarify this nomenclature.

2) How do the authors know they are recording from specific cell types? They state, “Cells were selected for recording using DIC optics under infrared illumination based upon somatic morphology and laminar position DIC optics under infrared illumination based upon somatic morphology and laminar position”. Current clamp traces are shown, however there are subtle variations of layer 4 fast spiking cells in the barrel cortex (Li and Huntsman 2014 Neuroscience). These variations include whether there is a delay in the first action potential and whether there is accommodation of action potential firing. The differences have biophysical significance in that rheobase and input resistance are different. The authors should provide intermediate traces to show these fast spiking cells are the same.

We are aware of this study and regret the oversight in citation. We acknowledge the diversity within the FS cell population within L4 and carefully he current-step induced firing patterns of our FS cell recordings before inclusion into this dataset. For both genotypes, we exclusively included FS cells that fired early in the injected current step and lacked a stuttering firing profile. We have added to the methods sections detailing this inclusion criterion, including a reference to the Li and Huntsman study and have added intermediate traces from current step injections for WT and Fmr1-KO FS cell recordings into the supplemental figures.

3) Is only looking at P10-11 timepoint sufficient? Would it be useful to look at older time points and younger with the same paradigms? How do the changes at this age P10/11 explain alterations seen at P14 or adult animals in Fmr1 Knock out animals? Please provide an explanation.

We chose to focus on P10-P11 for this study due to the ethological importance to activate sensation (onset of whisking), and circuit maturation (end of critical window for thalamocortical plasticity, onset of functional FFI) of this developmental milestone. Several studies have examined changes in

somatosensory circuit function in Fmr1-KO mice at more advanced stages of development (P14~P21, including Gibson et al., 2008, Antoine et al., 2018). Although some work has been performed on early second-postnatal week animals (e.g. Vislay et al., 2013, Gonçalves et al., 2013), detailed information about the circuit's frequency-dependent integration of thalamocortical input, evoked network response and spike output have not been provided. Whilst we feel that capturing a full dataset at another time-point is outside the scope of this study given the range of experiments performed, we confirmed key findings from previous work. Notably, although we report normal synaptic connection probability between pairs of L4 SCs at P10-11, by P14 this was found to be reduced in agreement with Gibson et al., 2008 (data not shown). We hypothesize that an initial developmental wave of synaptic connectivity emerges unperturbed between SCs in Fmr1-KO L4 despite the delayed critical period for thalamocortical plasticity (Harlow et al., 2010) as well as local circuit and cell-intrinsic effects reported in the present study. Further refinement of connections and their strength requires activity-dependent mechanisms (Ashby and Isaac 2011) sensitive to thalamically-evoked spike timing between pairs of SCs in L4 (Egger et al., 1999). Thus, we propose that several previously reported changes to network function in older Fmr1-KO animals could be reconciled with the circuit-level effects on L4 spike timing we report at P10-11 reported in this study. For example through affecting the maturation of SC-SC connectivity in L4 (P14, P21: Gibson et al., 2008), the maturation of L4-L2/3 excitatory connectivity (P14, P21: Bureau et al., 2008) and the correlation structure of population firing in somatosensory cortex (P14, P35: Gonçalves et al., 2013). Similarly, since maturation of connectivity between GABAergic and glutamatergic neurons in L4 is sensitive to normal sensory experience (Daw et al 2009, Chittajallu and Isaac, 2010), later-persisting changes to functional integration of GABAergic neurons in Fmr1-KOs (P14, P21: Gibson et al., 2008, P25: Paulszkewicz et al., 2011, P21, adult: Hays et al., 2011), may be reflective of earlier developmental insults reported here. We have added these arguments to the discussion section on the "Relevance to circuit-level defects observed in older Fmr1-KO animals".

4) The terminology is a little confusing. Do people commonly express these things as G/A ratio? Therefore also In/Ex is it commonly expressed this way? Most studies use E/I balance.

We have altered our nomenclature to better align with the published literature: Please see our response to Reviewer 2 on the same point.

Minor

1)The authors could really benefit by providing a summary figure.

We recognize the complexity of the breadth of experiments and analysis covered in this paper but feel that a summary figure isn't necessary given the experimental schematics provided for most of the figures. To clarify the

conclusions of the paper we have extensively edited the discussion, which we feel improves the presentation of our results and interpretation.

2) The reference manager that was used resulted in variations of the literature cited in the text.

We thank the reviewer for pointing out this discrepancy and have checked and corrected the irregularities in the cited literature.

Reviewers' Comments:

Reviewer #1:

Remarks to the Author:

The authors fully addressed all my concerns. This study is a major advance in the Fragile X field and, in my view, is acceptable for publication in the current form.

Reviewer #2:

Remarks to the Author:

The authors have responded positively to the criticisms and have improved the manuscript significantly. I have no more concerns.

Reviewer #3:

Remarks to the Author:

As mentioned in the previous review. This is an important paper for the field and worthy for publication in Nature.

The authors have adequately responded to reviewers concerns. The changes to the discussion have made the conclusions clear and succinct. This is an outstanding study.